# Behavioral evidence for the hierarchical execution of sequential movements
Darío Cuevas Rivera [1,2] ✉ & Stefan J. Kiebel [1,2]

Movements in humans and other animals are known to be hierarchically organized, with simple movements forming the building blocks to more complex, sequential movements, a phenomenon often referred to as chunking. Neuroimaging studies have highlighted this layered structure, implicating the primary motor cortex in simple movements, and pre-motor and parietal areas in sequences of movements. Behavioral experiments designed to study this hierarchy have required extensive training of simple movement sequences, such as key presses, using error rates and reaction times as indirect markers of chunking. In this work, we provide kinematic evidence that the hierarchical organization of movements naturally emerges during reaching movements toward large targets, without the need for extensive training. Through model-based analyses of the observed trajectories' geometry in a sequential pointing task (N = 20 participants), we infer the underlying hierarchy of the mechanism guiding movement generation. Our results show that most participants adapt their strategy dynamically using hierarchical planning, depending on the sequence. These findings offer insights into the process of chunking, as well as the conditions on how and when humans switch between flat and hierarchical planning during movement.

Body movements are known to be hierarchically controlled in the brain. Evidence for such a hierarchy of controllers has been found in neuroimaging and lesion studies of animals and humans, where simple, often short and single-target movements have been found to be learned and executed within the primary motor cortex[1–3], with recent evidence for pre-ordering of movements in the hippocampus[4], while the execution of complex movements has been found in pre-motor and parietal areas[1,5], and learning in basal ganglia[6] and the supplementary motor area[7]. In this hierarchy, the simple controllers in the primary motor cortex could be deployed by pre-motor and parietal areas to create sequences of these single-target movements.

Chunking[8] has been proposed as a mechanism underlying, at least partially, the hierarchical organization of movements, whereby individual movements that are executed in a sequence many times are grouped together in the brain and become a single action[9]. As such, chunking could be the result of the flattening of the hierarchical controller that was originally needed to perform a sequential action, simplifying its further execution. Direct behavioral evidence has been found, for example, in key-pressing experiments, where error rates and movement initiation delays have been used to infer chunking[10,11], or in sequential reach movements using the fusion index[12], which measures the drop in movement speed as one target is reached, and the hand moves on to the next target.

Despite the plethora of evidence in neuroimaging studies in both humans and non-human animals, direct behavioral evidence of the hierarchical organization of movement in the brain has proven difficult to observe[13]. Instead, behavioral studies have focused on measures related to chunking[10,12,14] and to pre-planning[11,15,16]. However, how these measures are related to the hierarchical organization of movements is still unclear: for instance, chunking is thought of as simplifying the control hierarchy (i.e., flattening), but could also be the result of a fine-tuning of the parameters of this hierarchy, leading to skilled performance after training. Similarly, pre-planning could be associated with both a hierarchical and a flat control strategy. Behavioral studies typically rely on long training sessions[15], often across multiple days, to induce chunking[10,14]. Furthermore, these studies have used extensive training to observe their results, which has been shown to reduce reaction times, error rates, inter-movement intervals, and movement fluidity even for untrained sequences[16–19]. Because untrained sequences also show these improvements in performance, it is unclear how much the improvement can be attributed to practicing a sequence in general and chunking of that sequence in particular. Taken together, this body of work indicates that behavioral evidence for hierarchical control has so far relied primarily on indirect timing- and error-based measures that are strongly influenced by training. This makes it difficult to dissociate changes in planning structure from more general improvements in motor execution.

[1]Faculty of Psychology, Technische Universität Dresden, Dresden, Germany. [2]Centre for Tactile Internet with Human-in-the-Loop (CeTI), Dresden, Germany.
✉e-mail: dario.cuevas_rivera@tu-dresden.de

What is therefore missing is a way to probe hierarchical organization at the level of movement generation itself, without relying on long training protocols or reaction-time proxies.

In this work, we introduce a rapid and training-free approach to obtain behavioral evidence of the hierarchical structure in human reaching movements. Our model-based method does not rely on secondary behavioral measures, such as reaction times or smoothness, nor on comparing early and late trials. Instead, it infers the presence of hierarchical control based directly on the geometry of the produced trajectories. Importantly, we show that even without extensive training, there is readily accessible and clear evidence in behavior that many sequential movements are organized hierarchically. This opens a critical opportunity: by revealing spontaneous hierarchical organization in simple movements, our approach provides a behavioral window into how hierarchical movement plans are formed and adapted in real time. In doing so, it lays the groundwork for future work to investigate the dynamics and mechanisms of hierarchical planning without the confounding effects of extensive training or overlearned behavior.

Human participants performed sequences of reaching movements towards large target circles on a table. Because of the size of the targets, participants performed their sequential movements with a high degree of coarticulation, i.e., modified their movement based on the preceding or following movements. Similar to previous studies[20–22], coarticulation in our task happens at the kinematic level, specifically in the trajectory of the fingertip as it goes from one target circle to the next. Contrary to previous experiments[20–24], we designed an experiment with large targets to lower accuracy demands and foster coarticulation. This kinematic signature is critical for our approach, as it provides direct access to how sequential movements are structured and linked, revealing the underlying hierarchical planning strategies.

We used two quantitative measures of coarticulation in combination with computational models to differentiate between flat (i.e., non-hierarchical) and hierarchical planning, both reflecting a combination of anticipatory[25] and carry-over[26] coarticulation. To do this, we model the two planning strategies with two closely matched models. Both models are grounded in the same optimization framework, the stochastic optimal controller (SOC)[27], and use identical movement primitives based on via-point control. The key difference lies in their architecture: in the first, a flat model plans the full sequence as a single, unified trajectory; we refer to this model as the via-point stochastic optimal controller (vpSOC). In the second, a hierarchical model introduces a minimal and neurobiologically plausible control layer that dynamically links simpler movement segments. This extension allows the hierarchical model to flexibly compose multi-target sequences in real time without altering the underlying control principles; we call this model the hierarchical sequential controller (HiSeq). Importantly, both models can plan movements in their entirety and differ only in which elements are required before movement onset: the via-point stochastic optimal controller requires all targets and control parameters to be planned before movement onset, consistent with the concept of chunking[9] and competitive queuing[19]. In contrast, the hierarchical sequential model can start by planning the first target(s) and add more as needed, consistent with recent observations regarding planning horizons in a similar sequential task[22]. By holding the core control principles and simple movements constant across models, we ensure that observed differences in fit arise from the presence of hierarchical structure, and not simply from a richer set of low-level parameters.

Concisely, the main objective in this work is to show that a flat planning strategy does not fit all trajectories observed in our experiment. We therefore hypothesized that many participant sequential trajectories would show geometries that cannot be produced with the via-point stochastic optimal controller alone. Concisely, the types and magnitude of coarticulation would vary between segments of the same sequential movement, contrary to the assumptions of the non-hierarchical (i.e., flat) optimal control model. We further hypothesized that many of these trajectories would be consistent with a hierarchical planning strategy, given the hierarchical model's ability to mix movement segments with different properties.

Consistent with our hypotheses, we found many participant sequences in which coarticulation and smoothness were outside of the range that can be produced with the via-point stochastic optimal controller. We further showed that trajectories were consistent with a hierarchical strategy, where the first two targets are grouped into one movement, and the third one is reached individually, with its trajectory dynamically linked to the initial two-target movement. Further analyses revealed a link between the hierarchical nature of some of the observed trajectories and existing measures related to chunking, namely the fusion index.

Our methodology and results have three important implications for further studies. First, they allow detecting the differences between hierarchical and flat movement planning strategies, without relying on measures that require comparisons across hundreds of trials of training that might affect even untrained movements. Second, they allow for this classification even in tasks with continuous hand movements where error rates are practically zero, as is the case in tasks with low-accuracy constraints. Third, by connecting flat planning to chunking via the fusion index, our results could help establish the mechanisms behind the process of chunking. Together, these advantages enable the study of chunking in motor tasks in which participants can be highly proficient and produce smooth, highly coarticulated trajectories.

## Methods
### Experiment
Participants ($N = 20$) performed reaching movements while sitting at a table, with the starting position and targets marked on the table (see Fig. 1). A computer screen displayed written instructions at the beginning of the experiment, introducing participants to the task, the printout, the expected movements, and the auditory cues used throughout the experiment. All instructions, animations, and auditory cues were programmed using PsychoPy[28]. No demographic data was collected for this experiment.

As shown in Fig. 1A, the printout consisted of colored circles, marking the initial position for all movements (about 4.5 cm from the participant's chest), and all the potential targets. The targets were divided into primary (red and blue circles, 11 cm in diameter) and secondary targets (olive and pink on either side, 7 cm in diameter, and lilac on top, 9 cm in diameter).

The sizes of the circles were chosen as follows: (1) The primary targets (blue and red) were chosen to be the largest to lower demands for accuracy. Given the results from previous experiments[23,24], we expected this reduction in accuracy to result in higher coarticulation. The secondary targets were reduced in size to be able to fit the entire printout in a comfortable workspace in front of the participants, while maintaining separation between the different targets. The exception to this is the top secondary target, which is larger than the ones on the sides. The size of this top target was chosen to enable participants to complete the upward trajectory in a straight line, for trajectories including this secondary target (sequences 2 and 5). For example, in sequence 2 (start-blue-lilac-start), a straight-line trajectory could touch the right edge of the blue target and the left edge of the lilac target, then move back.

After reading the instructions and hearing the auditory cues once, a short training phase started in which participants had to reach for the blue target and back to the starting position, a total of five times. This phase was used to familiarize participants with the auditory signals used at the beginning and end of each movement. During this phase, if necessary, the experimenter helped the participant understand when to start the movement and when a movement had been marked as finished. We did this to further familiarize participants with the auditory cues.

The following auditory cues were used: A go signal marked the start of a trial, and participants were instructed to move their fingertip quickly to the target(s). Different auditory cues at the end of the trial signaled if the movement had been performed too quickly, too slowly, successfully (chime), or unsuccessfully (buzzer). An additional auditory cue (lookup) signaled the participant to look up at the screen for further instructions, when necessary (e.g., at the end of a block).

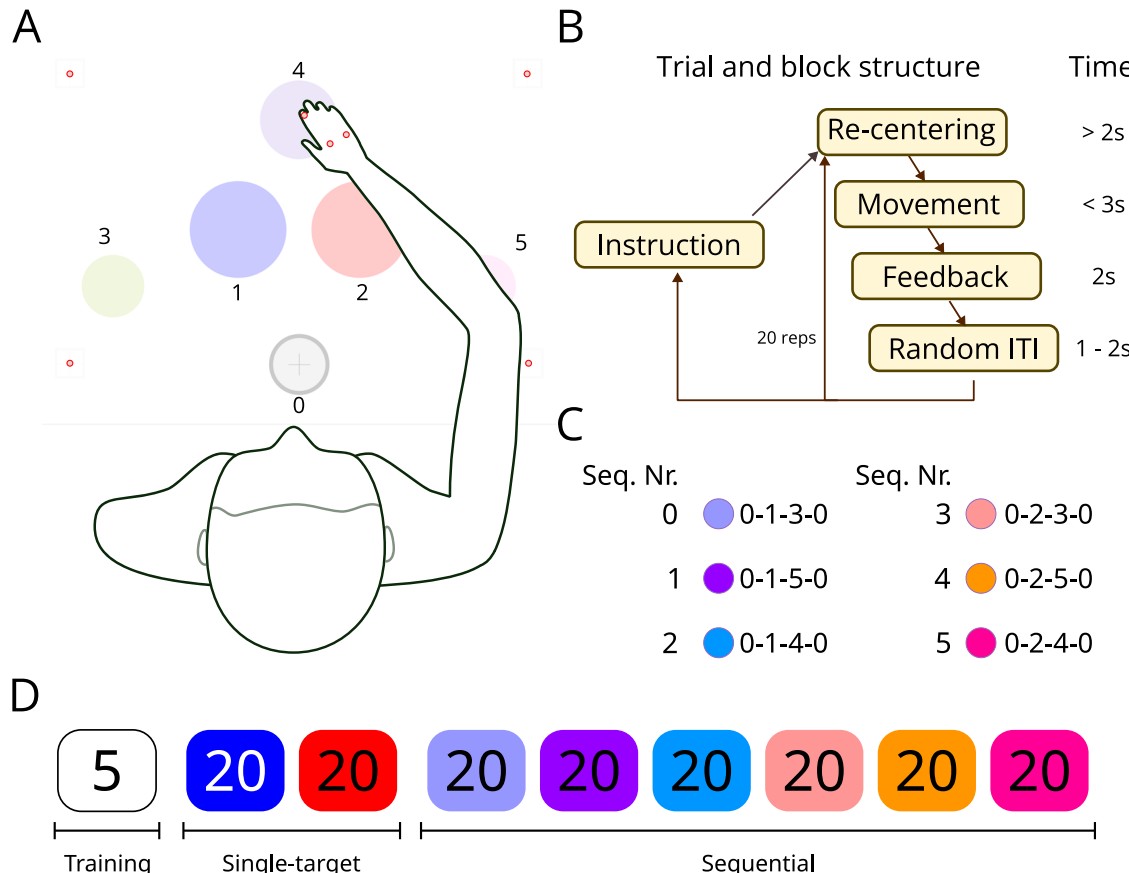

**Fig. 1 | Experimental design. A** Diagram of the experimental setup, in which a participant sits at a table with a printout of six colored targets. The targets are divided into starting position (0: gray on the diagram), primary targets (1: blue and 2: red), and secondary targets (3: olive, 4: lilac, 5: pink). These numbers are not visible to participants and are included here for clarity. The small red circles represent the positions of tracking spheres, including 3 on the participant's hand and four in the corners of the printout. **B** Trial and block structure. **C** Sequence numbering and color code, used for all figures. For each sequence number (e.g., 0), the target circles (e.g., 0-1-3-0) are shown in the order in which they must be reached, labeled as in **A**. **D** Order of blocks. Each rectangle represents a block with the same target (for single-target blocks) or a 3-target sequence (for sequential blocks). All experimental blocks had 20 trials except for an initial training block of 5 trials.

Movements were of two types: (1) single-target and (2) sequential. These movements were done in two phases, and each phase was divided into blocks of multiple repetitions of the same movement. The structure can be seen in Fig. 1D. During the single-target phase, participants had to reach for one of the primary targets (blue or red) and leave their hand there until an auditory cue signaled the end of the trial. During the sequential phase, participants had to reach for multiple targets in a sequence, always starting and ending on the initial position. An animation on the computer screen showed them the target(s) to reach; participants then placed their hand on the top (lilac) circle to start the block; this usage of the top circle as a "continue" button was consistent throughout the experiment, starting with the initial instructions.

Once a block started, the screen was turned off, and participants had to rely only on the auditory cues. Every block (single-target or sequential) consisted of 20 successful repetitions of the same movement; participants repeated the same movement until an auditory cue signaled the end of the block, so participants did not need to keep count of the trials. To begin a trial, participants had to place their hand on the starting position and wait for an auditory go-signal. After the signal, they had to reach for the target(s) within 3 s, or the trial would be considered a timeout (in which case they heard the timeout auditory signal). A trial was considered successful if and only if all targets were reached in the correct order, within the 3 s timeout, their hand remained in the last target for one second, and they kept an average speed between 600 and 900 mm s$^{-1}$. Unsuccessful trials did not count towards the 20 repetitions. Once a trial ended, participants had to move their hand back to the starting position, if it was not already there. Failure to do so would prompt the computer screen to display a reminder to do it. After a randomized intertrial interval, the next trial started with the go-signal. Once all 20 successful repetitions were completed, the lookup cue signaled participants to look up at the screen for further instructions.

The blocks were in the same order for all participants, starting with the single-target trials to the blue and then red targets, and then the sequential trials in the order shown in Fig. 1D.

Importantly, participants were explicitly instructed that to reach a target, it was sufficient to have their fingertip (of the index finger with the reflective marker) be located within the target circle, and that they did not need to reach for the center. The size of the target circles (between 7 and 11 cm in diameter) relaxed the accuracy requirements of the task, allowing participants to perform faster movements with more variability. We chose this design, which differs from typical sequential-movement experiments[23,29–31], to foster variability in trajectories, and to allow participants to generate coarticulated trajectories, i.e., trajectories in which the trajectory to a target is affected by the identity of the previous and the next target in a sequence.

Participants were instructed to take as many pauses as they wished. They could do this by taking their hand away from the table after a trial had ended. The prompt to return their hand to the starting position would show on the screen, and the experiment would be paused until the participant returned. At all times, the experimenter knew the sequence to be performed and could remind the participant in case they forgot, especially after a pause; participants were informed of this, but no participant in our experiment forgot the current sequence.

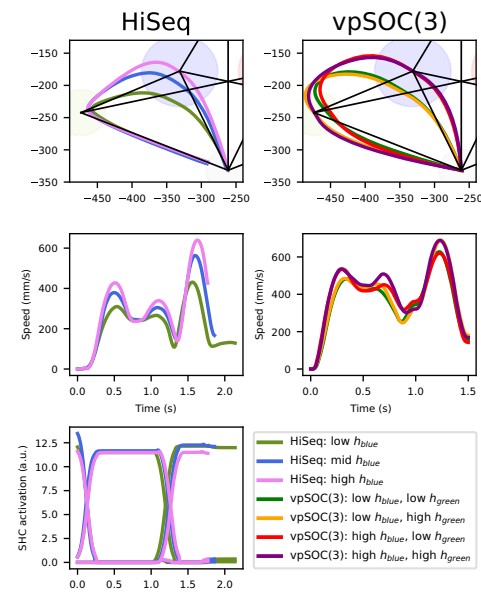

**Fig. 2 | Hierarchical sequential controller diagram and simulations.** The diagram shows the SHC space for Sequence 0 with 3 attractors ($\alpha_1$, $\alpha_2$, $\alpha_3$). Each attractor corresponds to a controller that drives the hand toward a physical target. As the trajectory moves in this space, the control matrices $L_t$ of the different segments are activated and mixed. The model begins movement from the starting position (gray circle) and reaches the first two targets (blue and olive circles) with a vpSOC(2) (controlled by $L^{(2)}$), as the SHC layer is moving towards $\alpha_2$. Close to the second target (the transition path indicated by the red arrow, representing the brief transition between $\alpha_2$ and $\alpha_3$), the control matrices are a dynamic combination of $L^{(2)}$ and $L^{(3)}$, creating transitional coarticulation, i.e., the rounding of the trajectory around the second target. The controller $L^{(3)}$ moves the hand back towards the starting position. Simulations are shown with different parametrizations of $h_{\text{blue}}$, which determines how close to the first target the trajectory will get. Speed profiles and the $z_i$ activations are shown. Similar simulations are shown with vpSOC(3).

The experiment lasted between 40 and 60 min, depending on the participant. All participants successfully completed all required movements, but the single-target movements for one participant (labeled 402) were missing due to technical problems with the data recording. This participant was not excluded from analyses, as all our analyses relied only on sequential movements. All participants gave written informed consent. The study was approved by the Ethical Committee of the Dresden University of Technology (SR-EK-189042023). The experiment was not pre-registered.

**Data acquisition.** The positions of the participant's palm and fingertip were tracked online in space throughout the experiment using a system with 11 infrared cameras (Qualisys AB, Gothenburg, Sweden), which relies on infrared cameras and three markers placed on the participant's hand to track with millimeter precision at a fixed 100 Hz polling rate. The position of the targets on the table was also tracked with four markers placed on the table (see Fig. 1A) to determine online when/whether targets had been reached, as well as to calculate the speed of movement.

**Speed profiles.** To show the speed profiles in the "Results" section, we split sequential trajectories into three segments, each one corresponding to one target. To find the points of separation between targets, we found the minimum speed of movement within each target, and split the trajectories into before-target and after-target based on the position of this minimum. We then normalized time for each segment, such that each segment had a duration of 1. The $x$-axis in the speed profiles in the "Results" section shows this normalized time.

**Models**

**Via-point stochastic optimal control (vpSOC).** We used the stochastic optimal control (SOC) model[27], widely used to describe the way in which the brain plans and executes movements in humans. This model consists of two parts: (1) A controller in the form $u_t = L_t \widehat{x}_t$, where $u_t$ is the control signal at time $t$ (i.e. the motor signal), $L_t$ is a time- and target-specific

control matrix and $\widehat{x}_t$ is the inferred hidden state at time $t$ (e.g. the hand position, inferred from noisy visual and proprioceptive information); (2) A state estimator based on a Kalman filter, given by $\widehat{x}_{t+1} = (A - BL_t)\widehat{x}_t + K_t(y_t - H\widehat{x}_t) + \eta_t$, where $A$, $B$ and $H$ are part of the linear model for the arm (see below). The matrices $L_t$ and $K_t$ are obtained by an iterative algorithm fully described and derived in Eqs. (4.2) and (5.2) of Todorov (2005)[27].

To adapt this model to sequential movements, we adopted an approach similar to that by Todorov et al. (2002)[32], in which a final target is set for the movement, with intermediary via-points that must be visited at set times during the trajectory. This was done via the $Q_t$ matrices, which encode a desired position of the plant at time $t$. For each target $i$ in the sequence, including the last one, a time $t_i$ must be chosen and the value of the matrix is set $Q_{t_i} = h_i^2 pp^\top + ff^\top + vv^\top$, where $f$ and $v$ are the costs of end-point speed and acceleration, adapted from Todorov (2005)[27] for longer movements. The free model parameter $h_i$ encodes the relative precision of the $i$th element of the sequence relative to the others and was used to generate the coarticulation observed in the data (see the "Results" section). Note that, contrary to the original implementation[32], we chose to make the $Q_t$ matrices for the via-points depend on the weights $f$ and $v$ in the same way as the final target does; see section 2, "Application to a 2D via-point" task in the Supplementary materials in ref. 32 for details. We chose this to make the $h_i$ parameters independent of the noise variances, simplifying interpretation of their values without affecting our results. We show sample simulations with the vpSOC(3) model for sequence 0 in Fig. 2, for different parametrizations of the model, showcasing its flexibility regarding halfway coarticulation and curvature.

In total, our implementation of the via-point stochastic optimal controller for 3-target sequences has 17 parameters, out of which 14 are fixed for all simulations (see the "Methods" section). Simulations applying the via-point stochastic optimal controller model to trajectories in the experiment can be seen in the Supplementary materials Figs. 1 and 3. For our results below, we simulated a total of 306 parameter combinations.

**Hierarchical sequential controller (HiSeq).** We developed a computational model that concatenates simple trajectories as modeled by the via-point stochastic optimal controller to construct sequences of movements, adapting the way one movement transitions into the next (coarticulation).

The hierarchical sequential controller consists of three layers. (1) The bottom layer consists of building blocks based on vpSOC(N) models for single-target and two-target movements. While in principle, three-target movements are part of the hierarchical sequential controller, all simulations with HiSeq were done without them. (2) The middle level consists of a dynamical system with internal dynamics that display trajectories that visit parts of the phase space in a given order (see below). (3) The top level establishes the sequence that the middle layer will perform. For a diagram illustrating the model (see Fig. 2). As can be seen, the hierarchical sequential controller is capable of controlling how close it gets to the first target without greatly affecting the trajectory around the second target, in contrast to vpSOC(3).

For the bottom layer, we instantiated internal models using the vpSOC(1) and vpSOC(2) components for each of the single-target and two-target movements, respectively. The time $t$ for each of the single-target movements was chosen to reflect the time requirements of the experimental task, i.e., around 700 ms; we scaled this time for the number of targets, equally spaced. While some participants in our experiment did not equally space their movements, we simulated other spacings (not shown) and found that making this simplification did not affect our conclusions in the "Results" section.

The middle layer is described by the generalized Lotka–Volterra equations given by

$$\dot{z} = \tau z(\alpha - \rho z) + \eta \tag{1}$$

where $z \in \mathcal{R}^{N_z}$, $\alpha \in \mathcal{R}^{N_z}$, $\rho \in \mathcal{R}^{N_z,N_z}$ and $N_z$ represents the number of target positions that the participant knows by instruction and $\eta$ is a zero-mean Gaussian noise term. $\tau$ is a time constant. These equations have one-hot equilibrium points $\widetilde{z}_i$, i.e., equilibrium points at positions where all dimensions are zero except one. It has been shown that the parameters $\alpha$ and $\rho$ can be set in such a way that the solutions to this system follow what is called a stable heteroclinic channel[33]: the solution travels from the vicinity of an equilibrium point to another, in an ongoing sequence that is reproducible under added noise and variability in the initial conditions.

The top layer controls the currently active sequential movement by changing the connectivity of the SHC layer, i.e., by changing the parameters $\alpha$ and $\rho$. At the beginning of each trial, a single sequence is active, and therefore, the top layer has no dynamics in our implementation.

When performing single-target movements, $z(t) = \widetilde{z}_i$ throughout the duration of the movement, for some $i$ that identifies the target. However, when a sequential movement is being executed, we modeled motor commands as a weighted sum of the commands of the vpSOC building blocks:

$$\widetilde{u}_t = \sum_i \frac{z_i}{\alpha_i} u_t^{(i)} \tag{2}$$

where $u_t^{(i)}$ is the motor command issued by the single- or two-target via-point stochastic optimal controller $i$. Because of the nature of the stable heteroclinic channels from Eq. (1), only one of the dimensions of $z$ is different from zero most of the time, with two being different from zero only at the transition from one target to the next.

The speed of the transition from one equilibrium point to the next can be controlled in a number of ways. If the initial conditions are set such that $z = \widetilde{z}_i$ for some $i$, then transitions would never occur, as, by definition, the trajectory will stay in the equilibrium point. Adding noise to the dynamics of the system in Eq. (1) (e.g., making $\eta > 0$) creates transitions, with higher noise levels creating faster transitions. We implemented a monitoring component within the SHC layer that sends a pulse of noise in the form of $\eta$ when the first sub-goal is achieved, e.g., when the hand is within the circle of

the current target. This noise term is not target- or sequence-specific. We show simulations with the model in Fig. 2 for different parametrizations, all using the vpSOC(2) + vpSOC(1) configuration. In this setup, the first two targets are reached through a single vpSOC instance, while the third segment is generated by a single-target vpSOC (equivalent to a SOC), concatenated by the SHS layer.

In total, our implementation of the hierarchical sequential controller for 3-target sequences has 20 parameters, out of which 17 are fixed for all simulations. Simulations applying the hierarchical sequential controller model to trajectories in the experiment can be seen in the Supplementary materials Figs. 2 and 3.

**Dynamical system of an arm.** The arm model is based on a discrete-time linear model that implements a low-pass filter, as used in the original SOC[27], as well as other model-based studies on motor control[34,35]. In this model, the state-space comprises the position (e.g., of the hand), its first two derivatives, the low-pass control, and a target dimension. The dynamics are given by

$$x_{t+1} = Ax + Bu_t + Cu_t + \epsilon_t \tag{3}$$

$$y_t = Hx + \varepsilon_t \tag{4}$$

where

$$x = [p, \dot{p}, f, g, \xi] \tag{5}$$

$$A = \begin{pmatrix} 1 & \Delta & 0 & 0 & 0 \\ 0 & 1 & \Delta/m & 0 & 0 \\ 0 & 0 & 1-\Delta/\tau_2 & 0 & \\ 0 & 0 & 0 & 1-\Delta/\tau_1 & 0 \\ 0 & 0 & 0 & 0 & 1 \end{pmatrix} \tag{6}$$

$$B = [0; 0; 0; \Delta/\tau_1; 0] \tag{7}$$

$$C = \sigma_c B \tag{8}$$

$$H = \begin{pmatrix} 1 & 0 & 0 & 0 & 0 \\ 0 & 1 & 0 & 0 & 0 \\ 0 & 0 & 1 & 0 & 0 \end{pmatrix} \tag{9}$$

where $\Delta$, $\tau_1$, $\tau_2$ and $\sigma_c$ are free parameters of the model, $\xi$ is the target, and $p, \dot{p}, f$ and $g$ are the position (e.g. hand position), speed, force and control filter, respectively. The observation matrix $H$ makes only the position, acceleration, and force observable to the agent.

For all simulations, the values of most parameters were chosen as in ref. 27, with exceptions described in their respective sections in the "Results" section. The default values can be found below.

**Coarticulation analyses**
We used two measures of coarticulation: (1) Halfway coarticulation and (2) transitional coarticulation. For halfway coarticulation, we calculated where the trajectory of a single trial crossed the halfway point of the center-to-center line that goes from the center of the initial position to the center of the first target (e.g., blue circle). To calculate this point, we drew a line orthogonal to the center-to-center line at its halfway point (see Fig. 3A). For each trial, the trajectory crossed this perpendicular line once (during the first segment); we call this a halfway cross (HC). Halfway coarticulation is the mean over the halfway crosses (HC), across trials per sequence (or target in single-target trials). HCs are positive if they lie on the "inside" of the movement, i.e., on the same side of the center-to-center as the second target.

For transitional coarticulation, we calculated where the trajectories crossed the line that bisects the angle between the center-to-center line from

**Fig. 3 | Measures of coarticulation. A** Halfway-point intersections of trajectories. The gray lines represent individual trajectories; the blue line is perpendicular to the straight line between the starting point and target, halfway between the centers of the starting position and the first target; each of the blue crosses represents the intersection of a trajectory with the blue line. The orange line is a representation of the distribution of the intersection. **B** Similar to **A**, showing where trajectories intersect the line that bisects the angle between the first leg of the sequential movement and the second.

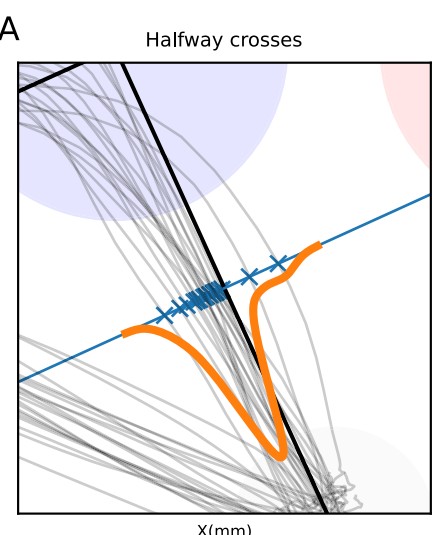

A
Halfway crosses

X(mm)

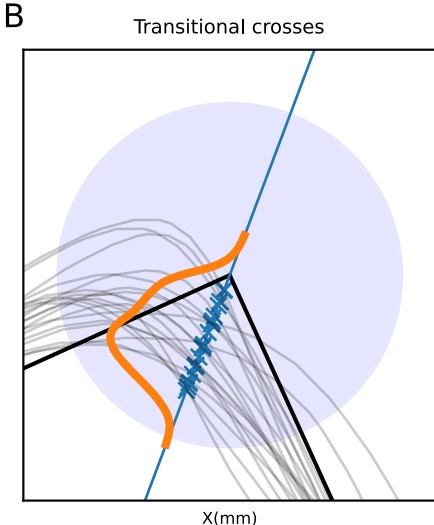

B
Transitional crosses

X(mm)

the initial position to first target, and the center-to-center line from first to second targets (see Fig. 3B). As before, we chose the sign of the HCs such that crosses on the side of the smallest angle between the two center-to-center lines, which also determines the "inside" of the trajectory.

### Curvature
The curvature of a parametric curve given by $x(t)$ and $y(t)$, where $x, y \in R$ is given by

$$C = \frac{x'y'' - y'x''}{\left((x')^2 + (y')^2\right)^{3/2}} \tag{10}$$

where $x'$ and $x''$ represent the first and second derivatives of $x(t)$ with respect to $t$.

### Analyses of significance
An important question is whether a trajectory is significantly coarticulated. In the "Results" section, we defined a trajectory as significantly coarticulated if the halfway crosses (HCs) had a distribution with a mean significantly different from zero. To establish this, we used the 20 repetitions of a single sequence to first determine if the HCs were normally distributed using the Shapiro–Wilk test for normality. If the samples were normally distributed, we tested if they had a mean significantly different from zero using a one-sample $t$-test; otherwise, we used the Wilcoxon signed-rank test.

Significance was in all cases evaluated at $p < 0.01$.

### Simulation parameters
Most parameters of the via-point stochastic optimal controller were set following Todorov (2005)[27], and their values were fixed throughout all simulations. Similarly, most parameters in the hierarchical sequential controller, including those regarding the Lotka–Volterra equations, were held constant throughout all simulations. The values of all relevant parameters can be seen in Table 1. The parameters marked with * are those from Todorov (2005)[27] that we did not explicitly mention in our brief description of the via-point stochastic optimal controller. Those marked with ** were changed depending on the simulation/figure. Their values for each one of the figures in the "Results" section can be found in the file paper_conf.py (see the section "Code availability", below), where the variable names in the table are used.

The number of parameters for vpSOC(3) is 17, and for the hierarchical sequential controller 20. However, most of these parameters are determined by the task, or were fixed identically across both models in all simulations. This includes the three parameters unique to the hierarchical sequential controller (monitor_distance, monitor_sd, and $\tau_{23}$), which were set to

match the target timing observed in participant trajectories. Note that vpSOC(N) can, in principle, include separate timing parameters for each segment, adding up to $N$ free parameters per sequence. In our simulations, however, we fixed these timings as equally spread (just as we fixed the timing-related parameters in the hierarchical sequential controller), so we did not treat them as free parameters in vpSOC(3).

What matters more than the total number of parameters of each model is how many parameters we varied to explain the behavioral data. The explanatory power of each model, as seen in the "Results" section, was obtained by changing three parameters for both models: $r$, $h_1$, and $h_2$.

### Fusion index and bootstrap analysis
The definition of the fusion index for a target $i$ is as follows[12]:

$$F = 1 - \frac{\frac{1}{2}(v_{\max}^{(1)} + v_{\max}^{(2)}) - v_{\min}}{\frac{1}{2}(v_{\max}^{(1)} + v_{\max}^{(2)})} \tag{11}$$

where $v_{\max}^1$ is the maximum velocity in the segment before the target, $v_{\max}^2$ is the maximum velocity in the segment after the target, and $v_{\min}$ is the minimum velocity within the target circle.

To test the significance of the differences between the two model-based categories, HiSeq and vpSOC(3), we used a bootstrapping approach. For all the participant-sequence combinations in the data that were categorized by the models (89 out of 120; see Table 2), we randomly assigned them into HiSeq or vpSOC(3), keeping the sizes of each category equal to the model-based ones. From this, we calculated the mean fusion index for each category and the difference between them. We repeated this proccess $10^5$ times and calculated how many of these $10^5$ differences were equal or larger than the one created by our model-based classification.

### Results
The goal of our study is to present behavioral evidence that supports the hierarchical structure of movement planning and execution of sequential reaching movements. To do this, we introduce two computational models: first, the stochastic optimal controller, a well-established model for the planning and execution of movements that does not rely on a hierarchy to plan sequences. Second, a model explicitly built on the idea that sequences are planned hierarchically. Crucially, these two models differ only in the presence or absence of the hierarchy, as both use optimal control theory to plan movements. By comparing the predictions from these two models to participant data, we show that many of the trajectories produced by our participants cannot be explained by the flat structure of optimal control alone, but instead require a hierarchical structure.

## Table 1 | Model parameters

| Parameter | Variable | Value | Description |
|---|---|---|---|
| $\sigma_c$ | sigma_c | 0.3 | Scale of the signal-dependent noise. Adapted to our experiment. |
| $m$ | m | 1 | Mass of the hand being controlled |
| $\omega_v$ | omega_nu | 0.4 | Cost assigned to speed discrepancies |
| $\omega_f$ | omega_f | 0.6 | Cost assigned to force discrepancies |
| $\tau_1$ | tau1 | 0.04 | Time constant for force |
| $\tau_2$ | tau2 | 0.04 | Time constant for acceleration |
| $\delta$ | delta | 0.01 | Time step |
| $\sigma_s$* | sigma_s | 0.5 | Scaling of sensory noise covariance |
| $\Omega_\omega$* | obs_noises | [0.02, 0.2, 1] | Covariance (diagonal) of observation variances, scaled by $\sigma_s$. |
| $\delta_t$ | delta_t | 0.01 | Time step, matching the polling rate of our motion capture system |
| $\xi_t$** | add_noise | 8000 or 10,000 | Additive noise in dynamics. This is set to zero in some simulations, denoted as such in the "Results" section |
| $r$** | r | – | Energy cost |
| $h_i$** | h_scaling | – | Cost assigned to missing the target. Alternatively called $h_{color}$ in the main text |
| $\tau$ | tau_23 | 3 | Time constant for the SHC |
| –** | monitor_distance | – | Distance to the target center (relative to the target radius) at which the monitor sends the signal to activate the next target |
| –** | monitor_sd | 5 or 10 | Amplitude of the signal from the monitor |

The "Results" subsections are structured as follows: (1) In the section "Data and simulation", we first describe the experimental data from our human participants, focusing on two representative participants. We then present behavioral analyses on all the participant data, focusing on elements that will be used in later sections to evidence hierarchical planning. (2) Finally, in the section, "Classifying trajectories as flat or hierarchical" we show the comparisons between the model predictions and the data.

### Data and simulations

We start by showing the data from two representative participants to visualize the phenomena that we will focus on in the following sections. Following this, we present multiple analyses of the trajectories observed in our experiment for all participants. The results from these analyses, as well as all characteristics of the observed trajectories discussed in this section, will be used in the following sections to infer the presence of hierarchically planned movements.

**Trajectories and speed profiles**. We chose two representative participants because they display two distinct and opposite patterns of coarticulation.

As a baseline, participants performed single-target reaching movements to each primary target. We show the trajectories of the fingertip marker in Fig. 4. As expected, single-target trajectories shown in Fig. 4 to targets 1 and 2 are mostly a straight line. Importantly, as can be seen in the Supplementary materials section "Participant data", these straight single-target movements were observed for all participants, showing that the deviations from straight lines observed in sequential movements cannot be explained by observation biases, nor from biomechanical constraints, and are instead signs of coarticulation.

These two participants display opposite trends in their sequential trajectories in Fig. 4 (Sequences 0–5), with participant A preferring more direct trajectories that always aim directly for the center of the target circles. Participant B, in contrast, showed more curved trajectories, i.e., movements that minimize hard turns. As we will show in the following sections, these two behaviors point towards different planning strategies for sequential movements. The data for all participants can be seen in Supplementary materials, section "Participant data".

As a participant moves their hand from target to target, the movement speed goes through local maxima and minima, where the maxima occur between targets and the minima around each target. As can be expected

from previous studies[23,36], for some trajectories, there are four distinct minima (the first and last being zero). This can be observed, e.g., in the trajectories for sequences 0 and 1; for others (those that go to the top target, i.e., sequences 2 and 5), the near-collinearity of the first and second targets can completely eliminate one of these minima, as is evident for participant B.

**Quantifying two measures of coarticulation**. In order to analyze the observed trajectories for all participants, we introduce here two measures of variance and deviation from straight lines in trajectories and show that they are evidence of coarticulation[25,26].

We first define a halfway cross as the difference between a trajectory going from targets $k$ and $k+1$, and the straight line that connects these two targets, at the halfway point between the centers of the targets (see Fig. 3A in the "Methods" section for more details). These deviations are a sign of coarticulation, as they depend not only on the current target, but also on the previous and next ones (if any); we therefore define the halfway coarticulation as the average of the halfway crosses (HC) over all 20 repetitions of the same sequence. This measure reflects a combination of anticipatory and carry-over coarticulation, in movements that have a preceding target and/or a following target, respectively. As each sequential trajectory contains multiple targets, we use the notation $hC^1$ to refer to the halfway coarticulation during the first segment.

As can be seen in Fig. 4, sequential trajectories differ greatly from straight lines that connect the centers of the targets in a sequence. To rule out that participants have a bias for curved trajectories in general, we tested whether the measured HCs in single-target movements had a mean significantly different from zero, separating movements towards the blue target (left) from those towards the red target (right), pooled for all participants. We found that HCs towards the left were not significantly different from zero ($t$-test, mean $= -0.72$ mm, SEM $= 0.34$, 95% CI $= [-1.41, 0.05]$, $t[379] = -2.1$, $p = 0.03$, $d = -0.1$), but movements to the right were ($t$-test, mean $= 5.57$ mm, SEM $= 0.25$, 95% CI $= [5.07, 6.08]$, $t[379] = 21$, $p \ll 0.001$, $d = 1.12$). However, the mean HC for right movements was small compared to the size of the target circles (110 mm) and compared to the length of the trajectories (about 170 mm). The pooled HCs towards the left and the right differed significantly from each other (independent two-sample $t$-test, mean[1] $= 0.72$ mm, SEM[1] $= 0.34$, mean[2] $= 5.57$ mm, SEM[2] $= 0.25$, 95% CI $= [-5.69, -4]$, $t = -11.27$, $p \ll 0.001$, $d = -0.81$). To see within-participant differences between movements to the right and to the left, we repeated these analyses on a per-participant basis. We found that for 10 out

**Table 2 | Model coverage of experimental data**

|  | HiSeq (%) | vpSOC(3) (%) | Both (%) |
|---|---|---|---|
| $hC^1$ range | 75 | 90 | 90 |
| $hC^1$ vs. curvature 2nd | 67 | 41 | 84 |
| Curvature 1st vs. curvature 2nd | 71 | 28 | 86 |
| Total | 48 | 9 | 69 |

The numbers are the percentage of participant–sequence combinations (averaged across trials) covered by each model (and both together), according to three tests. The row "Total" refers to all trajectories that meet the three criteria for each model, and the column "Both" refers to the trajectories that meet each criterion for at least one model. The intersection (bottom–right value) refers to all trajectories covered by a mixture of the two models.

of 19 participants, HCs for both sides differed significantly from zero (*t*-test, variable mean and SE, df = 19, $p \ll 0.001$, for each participant; see Supplementary Table 1 for all tests). For 8 participants, the HC of only one side was significantly different from zero, and for two, neither side. Finally, we found that for 15/19 participants, the left HCs were significantly different from the right HCs. Note that one participant had to be excluded from this analysis because their single-target data was corrupted.

In summary, we found that while single-target trajectories were generally different depending on the side of the target (left vs. right), the mean deviation from the center-to-center lines is small given the across-trajectory variance (which relates to motor noise) and the length of the movements.

The HCs for the two representative participants during the first segment of the sequential trajectories can be seen in Fig. 5A for each of the six sequences. We chose the sign of the HCs such that a positive HC is that in which the cross is on the same side of the center-to-center as the second target. For example, if a trajectory goes for the blue target and then the green (on the left), a positive HC would be one on the left of the center-to-center line between the initial position and the blue circle.

For the two representative participants, it can be seen that the HC distribution for single-target trials is centered around zero, but those for the sequential condition are not. To quantify this, we calculated, for each sequence and participant separately, whether their mean HC was significantly different from zero. We separated in this way because we found that both the sign and the size of the HCs varied within and across participants, making an analysis based on pooling insensitive to systematic differences. We found that the $hC^1$ of 81 trajectories out of the total of 120 (20 participants, 6 sequences per participant) were significantly different from zero (t-test, variable mean and SE, df = 19, $p \ll 0.001$ for each participant; see Supplementary Table 2 for all tests).

To further establish halfway coarticulation, we tested for a difference between HCs of single-target and sequential trajectories. Specifically, we tested for differences between trajectories on the same side (i.e., blue vs. red first target). We found that for all participants, at least 3 of the 6 sequences had HCs significantly different from the equivalent single-trial HCs (Welch's *t*-test, $p < 0.01$ for each participant-sequence combination; see Supplementary Table 3 for all tests).

In addition to halfway coarticulation, we found evidence of a second type of coarticulation as the movement transitioned from one target to the next. This can be observed as trajectories that do not go through the center of the target, instead having a distribution (over trials) that is centered elsewhere within the target circle. To analyze this, we determined for each trajectory separately the point at which it crossed a line that equally divides the angle between the center-to-center line to the first target and the center-to-center line from the first target to the second target (see Fig. 3B in the "Methods" section). We call these the transitional crosses (TC). We define the transitional coarticulation $tC^n$ as the mean of transitional crosses (TC) over all trials of the same sequence around target $n$. Like with halfway coarticulation, transitional coarticulation reflects both anticipatory effects of the upcoming target and carry-over effects of the preceding movement, but it is measured at the target rather than mid-movement between targets. As

can be seen in Fig. 5, the sequence-dependent distributions of the transitional crosses indeed depend on the following target. This difference is the most pronounced between sequences 0 and 2 for both representative participants. Because of this, we believe it is a measure of a mixture of anticipatory and carry-over coarticulation.

The TC distributions for each of the six 3-target sequences can be seen for the two representative participants in Fig. 5B. Each violin plot represents the distribution over 20 repetitions. As above, the positive sign is defined as that in which the trajectory is on the "inside" of the movement, i.e., on the smallest angle of the two center-to-center lines connecting the targets. As can be seen, transitional coarticulation is different from zero for most sequences for both representative participants. Indeed, we found, for each sequence and participant separately, that 76 out of 120 were significantly different from zero (one sample *t*-test, variable mean and SE, df = 19, $p < 0.01$; for 5 participant–sequence combinations, we performed a Wilcoxon signed rank test instead, as the repeated measures were not normally distributed according to a Shapiro test; see Supplementary Table 4 for all tests).

**Relationship between coarticulation and variability.** As a first step towards showing evidence for hierarchically planned movements, we show that the flat vpSOC(3) and hierarchical planning (HiSeq) model make different predictions regarding the relationship between halfway and transitional coarticulation. In these two models, two parameters have a strong effect on halfway- and transitional coarticulation: the cost of control $r$, which penalizes strong control signals, and the target precisions $h_i$, which determines the relative importance of reaching the center of each target $i$ in a sequence (see the "Methods" section). However, one important property of the via-point stochastic optimal controller is that the variability in trajectories and the types of coarticulation are inextricably linked. In sequential movements, $h_1$ (i.e., the precision of the first target) influences not only the variability of trajectories, but also the sign and magnitude of both types of coarticulation; this results in a strong correlation between variability, halfway and transitional coarticulation. The hierarchical sequential controller, in contrast, can disentangle the magnitude of transitional coarticulation from that of halfway coarticulation, as the former is heavily influenced by the SHC layer.

To compare the predictions from the two models with our participant data, we calculated the Pearson correlation between the mean of the HCs of the first segment of a sequence with the mean of the TCs around the first target of the sequence in participant data, separating by sequence and participant. We found no correlation between them (Pearson, $r(120) = -0.03$, 95% CI = $(-0.20, 1.15)$, $p = 0.74$). We found a weak correlation between the standard deviation of the HCs of the first segment and the mean of the TCs through the first target (Pearson, $r(120) = -0.19$, 95% CI = $(-0.17, 0.18)$, $p = 0.97$). Finally, we found a weak correlation between the mean of the TCs on the first target and their standard deviation (Pearson, $r(120) = -0.27$, 95% CI = $(0.11, 0.43)$, $p = 0.002$).

We performed the same correlation analyses on simulated trajectories generated by the vpSOC(3) model, varying coarticulation parameters. Contrary to what we observed in the participant data, we found (i) a weak correlation between $hC^1$ and $tC^1$ (Pearson, $r(1743) = 0.27$, 95% CI = $(0.22, 0.31)$, $p \ll 0.001$), (ii) a strong correlation between the standard deviation of the halfway crosses of the first segment and $tC^1$ (Pearson, $r(1732) = 0.49$, 95% CI = $(0.46, 0.53)$, $p \ll 0.001$) and (iii) a strong correlation between the standard deviation of the transitional crosses of the first target and $tC^1$ (Pearson, $r(317) = 7.26$, 95% CI = $-(0.67, 0.72)$, $p < 1e-50$). In contrast, the hierarchical sequential controller does not impose such a tight coupling between these features (see below), making it consistent with the weak correlations observed in the participant data.

These results point towards a hierarchical organization of movements when pooling all participants together. To further investigate these correlations, we analyze individual trajectories below to categorize them as either flat or hierarchical.

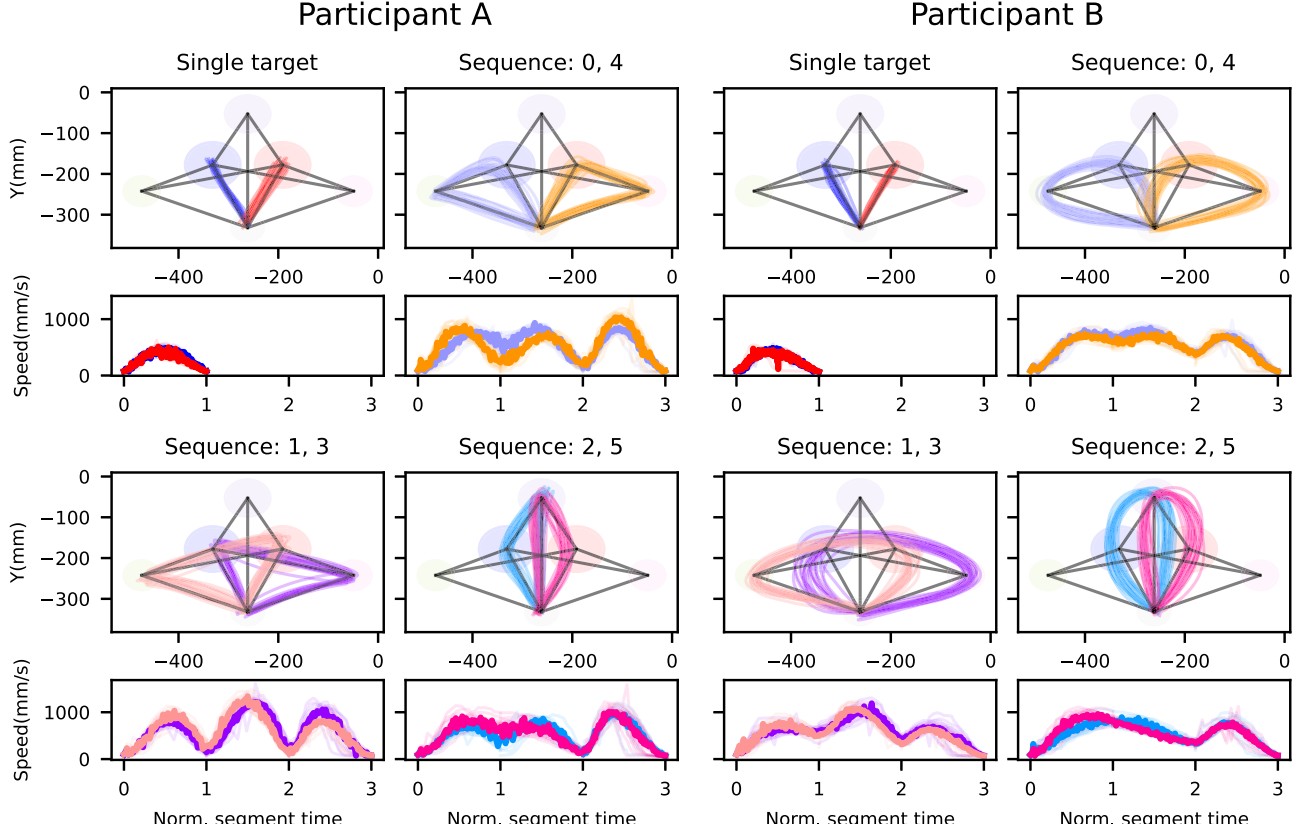

**Fig. 4 | Movement trajectories for two representative participants.** Participant A's trajectories are shown in the first two columns, and those of participant B in the last two. Each panel shows two different movements, and the corresponding speed profiles at the bottom. Each individual line represents one trial, using the sequence-based color coding (see Fig. 1C). Note that for the two single-target movements, the speed profiles overlap significantly, with the trajectories towards target 1 (blue) plotted behind the trajectories for target 2 (red). The black lines connect the centers of all relevant pairs of circles, for reference.

## Classifying trajectories as flat or hierarchical

We now build on the correlations from the previous section to categorize participants' movements as being consistent with either flat or hierarchical planning. To do this, we analyze the maximum curvature of trajectories (see the "Methods" section) as they transition from one target to the next. While lowering the maximum curvature around a target leads to coarticulation, it is not necessarily directly related to coarticulation. Indeed, as can be seen in Fig. 4, curvature and coarticulation are not correlated. Instead, maximum curvature is more closely related to segment fusion, which quantifies how continuously two movement segments are executed. This relationship is captured by the fusion index[12], where lower values correspond to higher curvature and thus more fused trajectories. Importantly, the usefulness of curvature as a measure will depend on the geometry of the task: if a sequence is to be performed with via-points lying on a straight line, maximum curvature will be near zero, while the fusion index could depend on the level of training the participant has. We will show that the hierarchical sequential controller's disentanglement of halfway and transitional coarticulation matches what we observed in participant data for many trajectories. Using these measures, we specify two criteria to determine whether a participant trajectory is compatible only with hierarchical planning (the hierarchical sequential controller), only with non-hierarchical planning (vpSOC(3)), or with both. This approach has two advantages: (1) it specifically relies on features that have been shown recently to be related to sequential movements and chunking[22]; (2) it directly tests whether each model can reproduce the observed combinations of coarticulation and curvature, establishing absolute limitations of their explanatory power.

The first classification criterion relies on a key difference between the two strategies: the hierarchical sequential controller, in contrast to

vpSOC(3), can produce trajectories that combine strong halfway coarticulation during the first segment with a sharp turn (i.e., high curvature) around the second target. This is because the transition around the second target is controlled by the mid SHC level, which operates independently of the lower-level movements (i.e., vpSOC(2)). In contrast, vpSOC(3) generates the entire sequence as a single trajectory, making $hC^1$ and the curvature around the second target inherently correlated in vpSOC(3).

The second classification criterion focuses on how the maximum curvature at targets relates to halfway coarticulation $hC^1$. In vpSOC(3), the maximum curvatures are largely determined by a single parameter, the cost of control $r$. As a result, the curvature around different targets tends to be tightly correlated: if the first turn is sharp, the second one will be as well. The hierarchical sequential controller, in contrast, decouples these curvatures. The curvature around the first target is set by the lower-level vpSOC(2) component, while the second is governed independently by the SHC dynamics. This flexibility allows the hierarchical sequential controller to produce trajectories where one turn is sharp, and the other is smooth—a pattern that vpSOC(3) cannot replicate.

For the first criterion, we used the maximum curvature of each trajectory within the area of its second target. In Fig. 6, we show this curvature for each participant, averaged across repetitions, separated by sequence. Circles represent individual trials and stars represent the average over all repetitions. To compare against what both models can produce, we simulated behavior with a wide range of parameter values, covering the widest range of behavior that the two models can produce. We show these simulations as black dots (for the vpSOC(3) simulations) and green rings (for the hierarchical sequential controller simulations). Each marker represents one model parametrization. The shaded areas, in the same color coding,

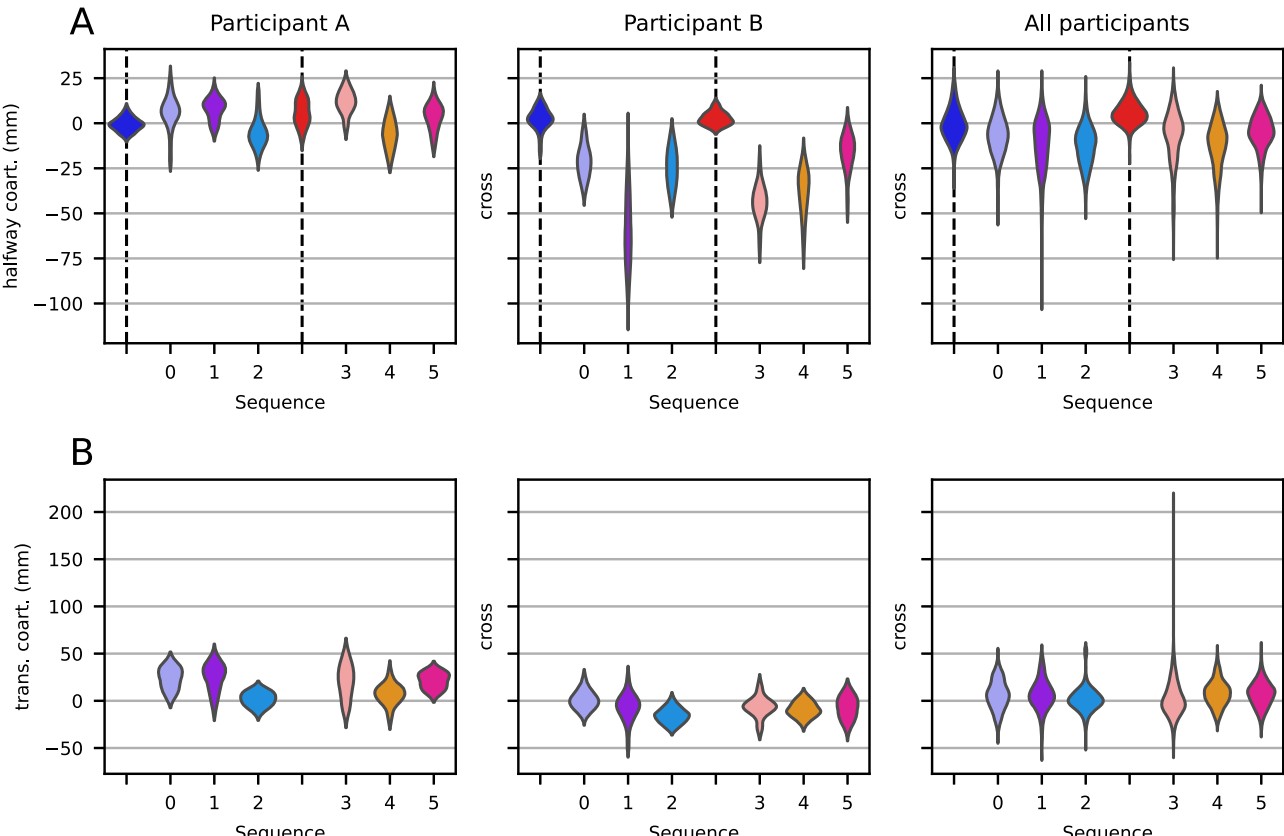

**Fig. 5 | Analyses of coarticulation for participants A and B. A** Halfway coarticulation ($hC^1$) for the first segment of all single-target and sequential movements for participants A and B ($n = 20$ for each violin), as well as for all participants pooled together ($n = 380$ for single-target and $n = 400$ for sequential, per violin), in the sequence-based color coding from Fig. 4. Sequences are labeled in the $x$-axis, while single-target movements are marked with a black dashed line. **B** Transitional coarticulation ($tC^1$) around the first target of the six 3-target sequences ($n = 20$ per violin for individual participants, $n = 400$ per violin for all participants).

represent the area of coverage for each model, interpolated from the observed simulations (see the "Methods" section).

As can be clearly seen in Fig. 6, the hierarchical sequential controller covers a much wider area than vpSOC(3), including more of our participant data: For instance, multiple participant–sequence combinations show a much higher curvature for highly negatively coarticulated trajectories (i.e., those between $x$-values of $-20$ to $-10$ within the green area but not the black area). This greater coverage is not a trivial consequence of model complexity: both models were simulated by varying the same three parameters. Thus, the additional flexibility of the hierarchical sequential controller reflects its hierarchical structure rather than free parameterization. To ensure that this comparison does not bias results toward the hierarchical sequential controller, we combine the coverage analysis with a significance test: trajectories are considered incompatible with vpSOC(3) only when they lie outside its coverage and show coarticulation significantly different from zero. This controls for motor noise and confirms that the flat planning model has fundamental limitations. Importantly, the simulations were obtained by varying the values of three parameters for both models; therefore, the extended area of coverage of the hierarchical sequential controller cannot be explained by an increased model complexity. Because of this, the flexibility seen in these trajectories can only be explained by the hierarchical structure of the hierarchical sequential controller. This analysis could bias the results in favor of the more complex hierarchical sequential controller model. However, it is important to note that we combined the area-of-coverage comparison with a significance test to conclude that some trajectories cannot possibly be modeled with vpSOC(3), highlighting that its flat planning approach is limited in its scope. Importantly, the significance test allowed us to account for variability in the trajectories caused by noise in the motor signals.

For the second criterion, we calculated the maximum curvature around the first vs. second targets for participant data, as well as for the simulations with vpSOC(3) and the hierarchical sequential controller from Fig. 6. In Fig. 7, it can be seen that vpSOC(3) simulations (in black) show strong correlations between these two curvatures, i.e. the black dots representing vpSOC(3) simulations span a very narrow range of $y$-values (2nd target curvature) for any $x$-value (1st target curvature), and the higher the curvature around the 1st target, the higher that around the 2nd target. Most of the variability in curvature for both targets in vpSOC(3) is due to the parameter $r$: lower values of $r$ create higher curvature for both targets, and variations in other parameters create clusters with small changes in curvature.

In Fig. 7, it can be seen that the variability between different parametrizations of vpSOC(3) does not cover the range seen in the participant data. In contrast, the hierarchical sequential controller can disentangle the two curvatures and is therefore able to cover a much wider range of the data, including areas where one curvature is low and the other high. This coverage can be seen in Fig. 7 (green shaded area), where individual trials in participant data are shown as small circles (color-coded by sequence), and the average across all 20 repetitions as stars. As can be seen in each panel, most individual trials, as well as averages, are within the green-shaded area, and many are outside the gray-shaded area of the vpSOC(3) coverage.

To quantify how well each model explains the observed trajectories, we assessed the proportion of trials covered by their respective prediction spaces in Figs. 6 and 7. We accounted for trial-by-trial variability by calculating the 99% confidence intervals for the mean of each star and determined whether it intersected each model's coverage. The results are summarized in Table 2, which includes not only the classification criteria from Figs. 6 and 7, but also coverage based on the maximum (negative) $hC^1$.

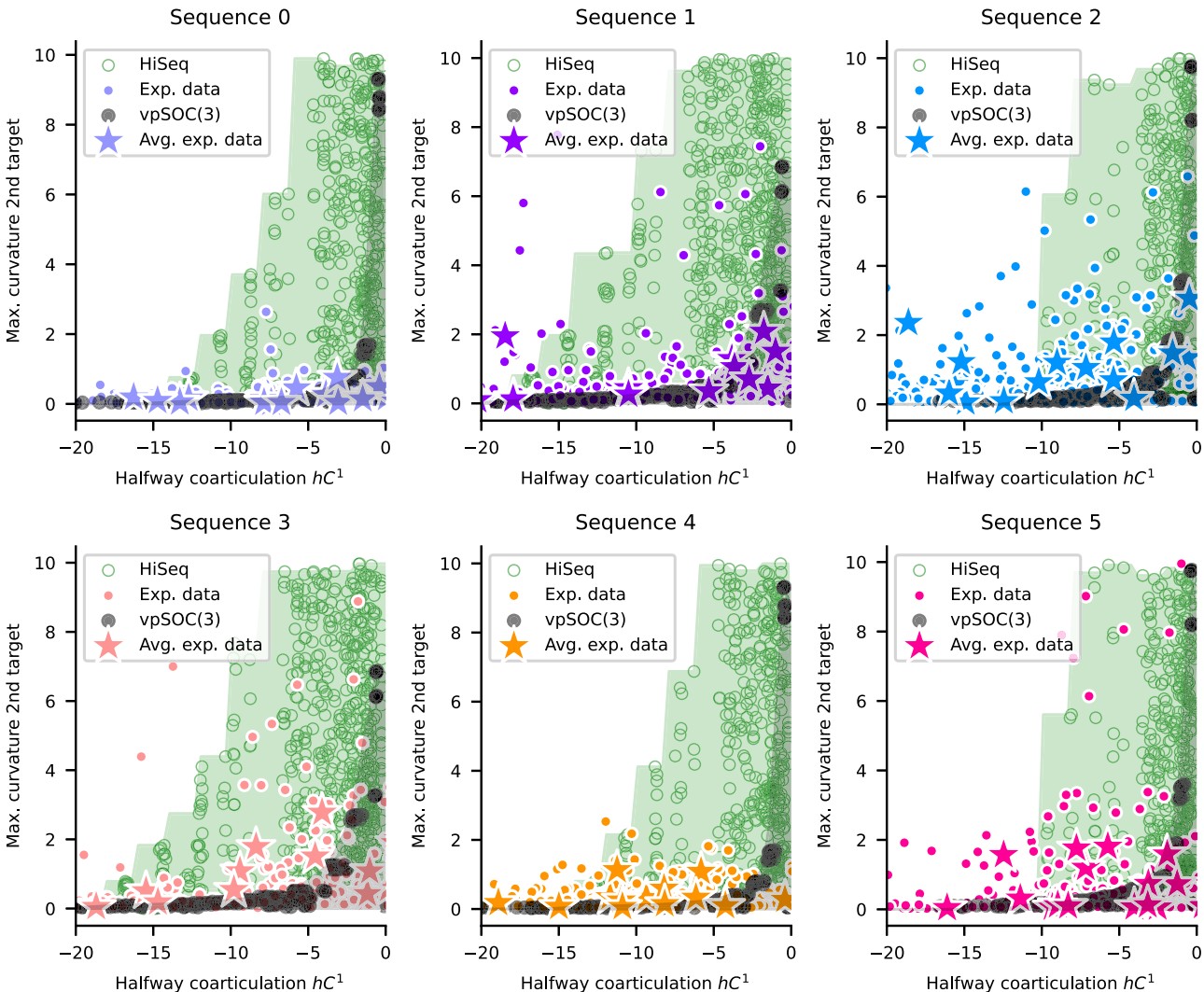

**Fig. 6 | Model coverage of curvature and halfway coarticulation, separated by sequence.** For each panel, the $hC^1$ ($x$-axis) is plotted against the maximum curvature around the second target ($y$-axis). The color coding is as in Fig. 1C. Each colored dot represents one trial for one participant. Stars represent the average over all trials for one participant. Only those with negative halfway coarticulation are shown ($n = \{301, 326, 360, 277, 336, 274\}$ for each sequence). The shaded areas (black and green) are those covered by vpSOC(3) and HiSeq, respectively ($n = 306$ and $n = 1008$ simulations, respectively, cropped to data). Note that the green-shaded area includes most of the black-shaded area.

Overall, neither model alone explains the majority of participant behavior: the hierarchical sequential controller accounts for up to 49%, and vpSOC(3) only 9%. However, a version of the hierarchical sequential controller that incorporates both vpSOC(2) + vpSOC(1) and vpSOC(3) trajectories would account for 69% of the observed variability.

Taken together, our results show strong behavioral evidence for hierarchical planning in at least 48% of trajectories in our participant data.

Given the relation between the hierarchical organization of movements with chunking, an important question is whether our model-based classification is related to existing measures of chunking. Recently, the fusion index was proposed as a measure of chunking[12]. The fusion index reflects the difference between the minimum speed around an intermediary target and the maximum speeds of the segments that surround it (see the "Methods" section for the mathematical definition). The fusion index is in the interval [0, 1]; a value of 1 means the trajectories are fused, i.e., all targets are reached in one swipe. A value of 0 means the trajectories are performed in segments.

We first show in Fig. 8A that halfway coarticulation $hC^1$ is highly correlated to the fusion index, similar to the findings by Sporn et al. (2022)[12]. This correlation does not hold for trajectories with positive $hC^1$, which is likely due to the geometry in the task, which does not allow very high values

of positive $hC^1$, as well as the low number of positively coarticulated trajectories for some sequences, with sequence 2 having none.

We expected our classification of trajectories to lead to vpSOC(3)-like trajectories having a higher fusion index than HiSeq-like trajectories. In Fig. 8B we show the distribution of fusion index values for both models. As expected, the mean fusion index (green triangles) across trajectories for both segments is higher for vpSOC(3) than for the hierarchical sequential controller. To rule out the possibility that these differences between groups were not due to random chance, we performed a bootstrap analysis (see the "Methods" section for details) and found a probability $p < 0.01$ of finding a difference between groups as large or larger than that of our model-based classification.

## Discussion
In this work, we demonstrated evidence for the hierarchical organization of movements in a sequential reaching task, using a model-based approach to analyze the geometry of the movements produced by 20 human participants. We used an existing model for human motor control, namely the stochastic via-point optimal controller[27] (vpSOC), and introduced a hierarchical version (HiSeq) based on the same via-point stochastic optimal controller but with an additional lightweight hierarchy on top to

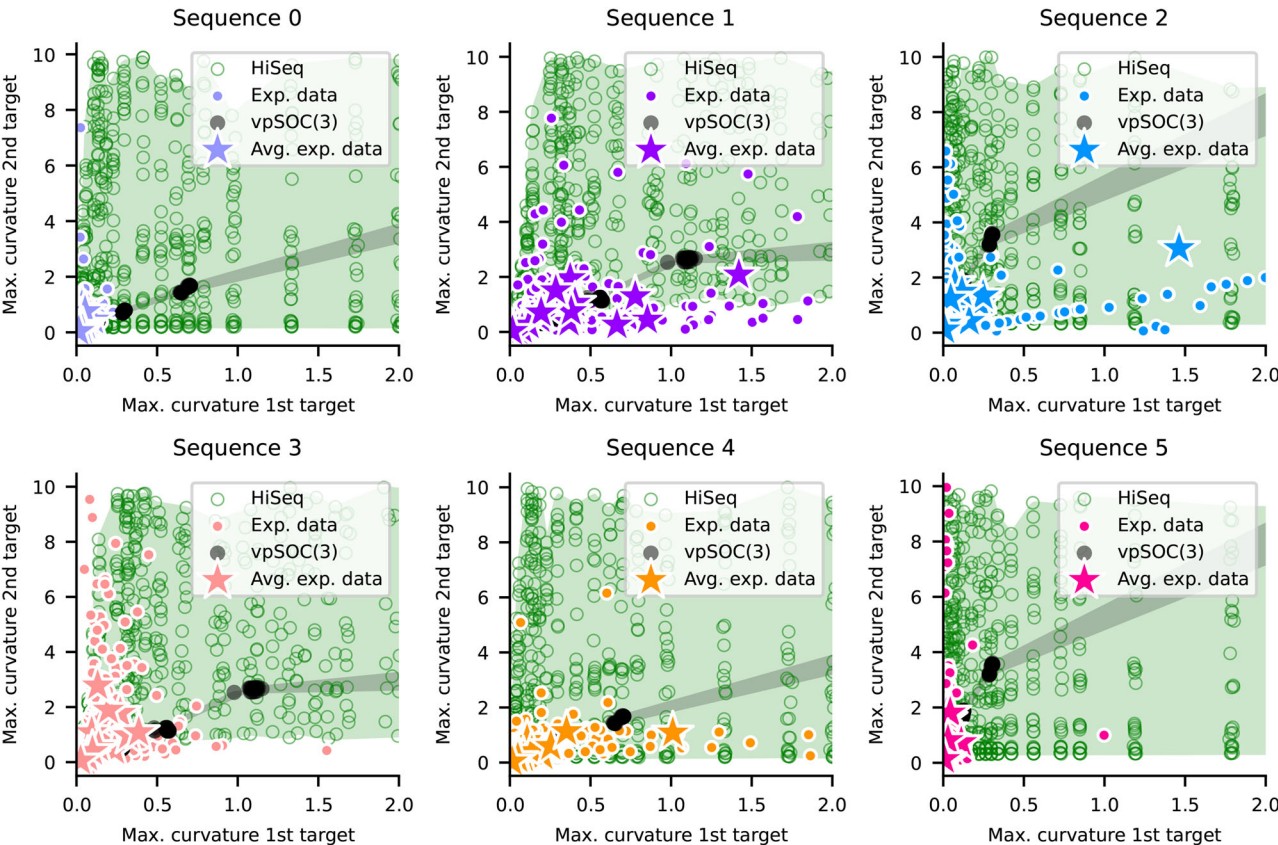

**Fig. 7 | Model coverage of curvatures.** Maximum curvature around the first (*x*-axis) plotted against the maximum curvature around the second (*y*-axis) target, one panel for each sequence, for participant data (sequence-colored circles; *n* = 400 per panel)), for the simulations with vpSOC(3) (black stars; *n* = 306 simulations, cropped to data) and with HiSeq (green circles; *n* = 1008 simulations, cropped to data). Same simulations as in Fig. 6. Shaded areas visualize the coverage of HiSeq (green) and vpSOC(3) (gray). The axes were cropped to values calculated for participant data, except for sequence 2, where outliers were cropped for clarity (discussed in the main text).

concatenate simple movements into sequential trajectories. Using a sequential movement experiment where participants reached for targets in a sequence, we identified distinct trajectory features of human sequential movements that could not be reproduced by the flat model but were captured by the hierarchical model.

Importantly, and in contrast to standard experiments, we designed the task using large target circles to foster coarticulation at the kinematic level. This enabled us to observe and directly model coarticulation. Coarticulation itself can be seen as a sign of planning ahead, as it can be predictive of the next target in a sequence, as is the case in our experiment. However, as both planning strategies (flat and hierarchical) can produce trajectories with coarticulation predictive of the next target, coarticulation is not, by itself, sufficient to infer a hierarchical controller. To account for this, we focused on the sign of coarticulation and the maximum curvature as trajectories transitioned from one target to the next. The two features, coarticulation and curvature, are important because their relationship reveals how the movement was planned. If the entire sequence is generated as a single movement (flat planning), these two features are tightly linked. But if the sequence is composed of dynamically linking simpler movements (hierarchical planning), they can vary more independently.

Consequently, using our model-based analyses, we found that about 48% of our observed trajectories showed evidence of hierarchical planning. We found that individual participants showed a mixture of strategies, with some sequences showing evidence of hierarchical planning and others of flat planning, similar to previous findings[22,24].

Our simulations and the results in Figs. 6 and 7 were obtained with a HiSeq model with a vpSOC(2) for the first two targets. We chose this because we found the complement configuration (i.e., vpSOC(1) + vpSOC(2)) to provide a poor match to the observed halfway coarticulation on the second segment, and the curvatures around both targets. From these results, we inferred that those movements that are hierarchical must be dynamically linked at the second target, consistent with recent findings in a similar behavioral task[22].

### Limitations

One limitation of our analyses is that the combined models do not account for the full variability observed in participant data. For example, the magnitude of the coarticulation observed in at least 10% of participants is higher than what either model can produce. This limitation is not necessarily due to the models themselves, but to their parametrization. In particular, parameters such as the mass and noise of the plant were set as described by Todorov (2005)[27]. In additional simulations (not shown) we found that more participant data could be explained by, for example, increasing the magnitude of halfway coarticulation by increasing the mass. However, it goes beyond the scope of this study to explore the limits of these models of muscle activation. Another potential improvement could be achieved by the simulation of the entire arm with its degrees of freedom; however, due to the non-linear dynamics of such a plant, a different derivation of the SOC model would be required, for example, by deriving a controller and estimator specific to the plant[37], or by using extended or unscented Kalman filters as estimators[38].

Previous approaches have shown that even simpler models than SOC can be fitted very closely to hand-movement trajectories during sequential reaching tasks[20,21,37]. In this work, we chose not to perform model fitting, instead focusing on a method by exclusion, with which we showed that some trajectories cannot possibly be modeled with vpSOC(3) (see Table 2). Model

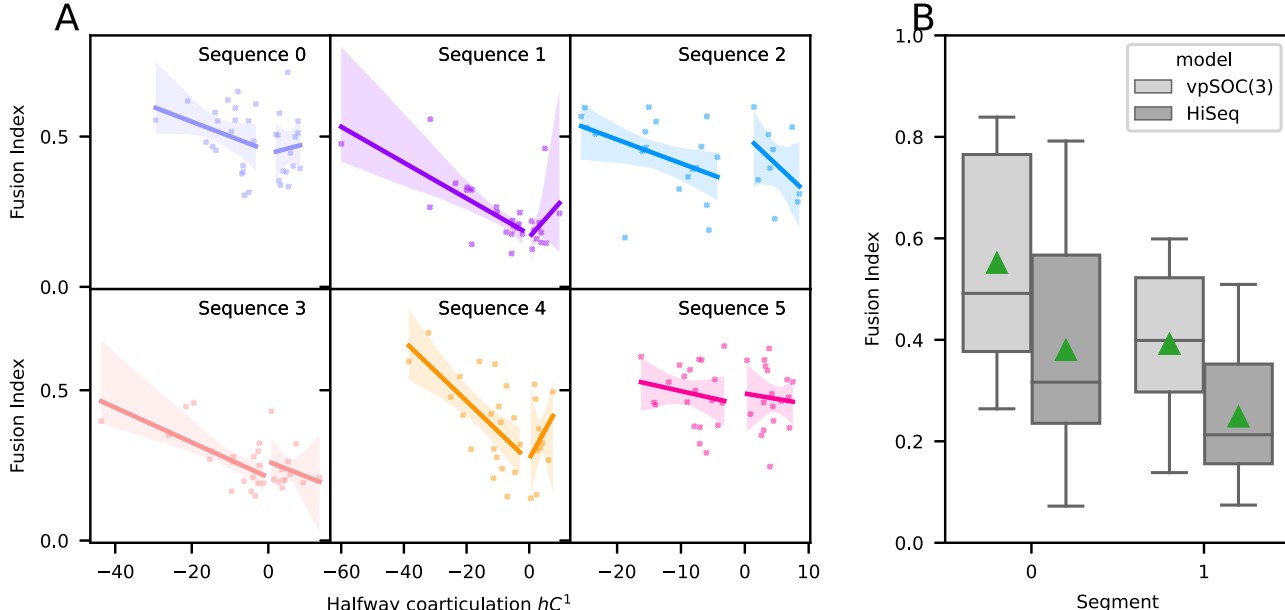

**Fig. 8 | Fusion index and coarticulation. A** Relation between the fusion index and coarticulation, separated by sequence (different panels) and sign of the halfway coarticulation (colors). Each cross represents one participant–sequence combination ($n = 20$ per panel). Lines represent best-fitting linear regression, with shaded areas representing 95% intervals for fitting. Fitting was performed separately for negative and positive coarticulation. **B** Fusion index per model and segment. Green triangles represent the mean. $n = 120$ for each box plot.

fitting would not necessarily provide further insights in this regard, but it could have important uses in future works. For instance, an important advantage of model fitting is that it enables the use of the fitted models for the prediction of future movements. Furthermore, it would in principle enable the search for neural correlates of the model variables in neuroimaging studies[1,39].

Finally, our main result is to show that a flat controller is not capable of displaying the wealth of trajectories observed in our experiment. We did this by showing that the state-of-the-art model based on optimal control could not reproduce many of the trajectories. We introduced the hierarchical sequential controller as the simplest extension of the via-point stochastic optimal controller to a hierarchical controller that is capable of not only linking individual components (e.g., vpSOC(2) and vpSOC(1)), but also of creating coarticulation at the transition between the two. The choice of the generalized Lotka–Volterra equations as the top layer was based on their previous use in neuronal dynamics[40,41], but we did not specifically test their viability in this application.

**Alternative formulations**

Before discussing specific model components, it is important to clarify what we mean by hierarchical planning. We define it operationally as the presence of a slower, higher-level process that dynamically links simpler movements into a sequence. A key question is whether flat models could reproduce the observed dissociation between halfway and transitional coarticulation without introducing an equivalent hierarchical mechanism. As we argue below, matching our experimental findings would, in practice, require incorporating some form of hierarchical control operating at a slower timescale.

An important point to address is our choice of components in the hierarchical sequential controller, namely SHCs and the via-point stochastic optimal controller, as well as the use of the latter as the representative for optimal controllers and as the mechanism for non-hierarchical sequential planning. Indeed, since the introduction of the SOC, and an earlier introduction of deterministic optimal controllers as models for human movement[32,42], multiple models have been derived from the same basic principles: many of them rely on the same or very similar controllers as SOC,

focusing instead on the sensory feedback components[30,43] and would therefore produce similar results to the via-point stochastic controller under the conditions of our experiment. Other models cannot be directly applied to sequential movements due to a lack of a fixed, explicit representation of time[34,44]. Yet others, such as Guigon et al. (2019)[45], already contain a hierarchy similar to the hierarchical sequential controller, whereby another agent (in their case, the experimenters) chooses a set of intermediate targets for the motion to follow. In those studies, however, these targets were not explicitly defined by the task or the environment; instead, they were hand-selected at very short intervals to match the fluctuating speeds of hand movements observed in their experiment. Attractor-based models[46] could be expanded using a hierarchical system to implement sequences of targets. Such models would, however, fail to reproduce negative halfway coarticulation, as they lack the multi-target planning that the via-point stochastic optimal controller has. This leaves us with only two main model families, which could be an alternative to hierarchical planning: those based on stochastic controllers (like SOC), and those using deterministic controllers. While our results were obtained using the stochastic formulation, the underlying principles, particularly the role of hierarchical sequencing, should generalize to deterministic variants as well.

The hierarchical sequential controller relies on a monitoring component in the SHC layer for transitioning from one movement to the next. This monitor tracks how close a movement was to completion and preemptively activates the next one. This implementation has the advantage of adapting to the size of the target and presents the possibility of implementing a speed-accuracy trade-off, which, similarly to Fitts' law, could be affected by the difficulty of the task, as was observed in earlier experiments[24]. Practicing a sequence could, as a consequence, shift this monitoring system to, for example, a timing-based transition between targets, where the timing becomes part of the movement program[47]. A similar idea has been developed in robotics[48], and further control over the exact timings of the elements in the sequence was investigated by Horchler et al. (2015)[49].

As shown in Figs. 6 and 7, the hierarchical sequential controller covers an area in the curvature and halfway coarticulation spaces that is wider than that observed in our experimental data. This is due partially to our choice of plotting all simulated trajectories for all model parameter values in every

**Article**

panel for completeness. In addition, the hierarchical sequential controller and, to a lesser extent, vpSOC(N), can generate trajectories that were not observed in our study. With our sample of trajectories from 20 participants, we cannot determine whether these unobserved trajectories are also part of the broader human repertoire. In principle, a Bayesian approach could be used in a future study to limit these models' flexibility to those observed in a significant sample of the general population: by using Bayesian model fitting techniques, a larger data set could be used to obtain posteriors over parameters, which could subsequently be used as priors for the entire population.

## Skill learning and chunking

Our monitoring-based implementation does not preclude the learning of expected timings for the completion of each individual movement. A combination of timing and monitoring could enable the agent to generate movements with reduced reliance on sensory feedback, while still benefiting from it.

In this perspective, the consolidation of a sequential movement through training not only sets expected timings for each segment of the sequence, but also turns the sequential movement into one more action that the agent has at its disposal, a process often called chunking[10,40,50]. In other words, training could transform a HiSeq movement into a vpSOC(3) movement, changing the way this movement is performed. Indeed, we implicitly used chunking when defining the hierarchical sequential controller, whereby a two-target sequence can be controlled directly with a vpSOC(2) model. Such a controller, with a sequence of two targets, could have arisen from the consolidation of a HiSeq-like movement with two separate targets, and eventually turned into a single memory unit[10]. In this way, chunking can be seen as the creation of a deeper hierarchy of movements: controllers (either SOC-like or SHC-based) manage a hierarchy of concatenations that starts at the simplest reaching sequences, e.g. creating a two-target sequence (1,2), and moves up in complexity and duration, e.g. with sequences ((1, 2), 3) and (((1, 2), 3), 4), similar to those observed by Martins et al. (2019)[51] with finger tapping.

## Conclusions and outlook

In this work, we focused on finding behavioral evidence for a hierarchical planning strategy for sequential movements (HiSeq). Having found this evidence, an important question remains: under which circumstances do humans use hierarchical movement planning? We speculate that movement difficulty, expressed in terms of diminishing target size or increasing speed requirements (i.e., Fitts' law), could trigger the usage of hierarchical movements. Indirect evidence for this has been observed[24], where it was found that the effects of a second target on the trajectory towards the first (i.e., anticipatory coarticulation) could be suppressed by decreasing the size of the first target. This could be interpreted as participants concatenating the second part of the movement after the first had ended, only when the first target was small. Additional difficulty considerations, such as the number and size of the required muscles, as well as coordination requirements (e.g., performing sequences of hand gestures), could make planning with the via-point stochastic optimal controller computationally too expensive. The sequence length, i.e., how many movements must be performed after each other, could also play an important role in choosing a planning strategy. Finding evidence for the effects of these task characteristics on planning strategy could be the focus of future experimental studies.

## Data availability

The raw behavioral data were published in an online repository (link to repository)[52] in the Motion-BIDS standard[53].

## Code availability

The code used to generate the figures and results in this work can be found in an online repository (link to repository)[54]. Instructions to set up a Python environment to use the code are contained within the README.md file. The pre-processed behavioral data and simulated data used for the results can also be found in this repository.

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

## Acknowledgements
Funded by the German Research Foundation (DFG, Deutsche Forschungsgemeinschaft) as part of Germany's Excellence Strategy—EXC 2050/1—Project ID 390696704—Cluster of Excellence "Centre for Tactile Internet with Human-in-the-Loop" (CeTI) of Technische Universität Dresden. The funders had no role in study design, data collection and analysis, decision to publish, or preparation of the manuscript.

## Author contributions
Dario Cuevas Rivera: experimental design and measurement, simulations, analysis design and programming, manuscript drafting and revision. Stefan Kiebel: analysis design, manuscript drafting, and revision.

## Funding

## Competing interests
The authors declare no competing interests.
