## [Transparent Peer Review file · Communications Psychology]

Behavioral evidence for the hierarchical execution of sequential movements

Corresponding Author: Dr Dario Cuevas Rivera

Version 0:

Decision Letter:

Dear Dr Cuevas Rivera,

Thank you for your patience during the peer-review process. Your manuscript titled "Behavioral evidence for the hierarchical execution of sequential movements" has now been seen by 3 reviewers, and I include their comments at the end of this message. They find your work of interest but raised some important points. We are interested in the possibility of publishing your study in Communications Psychology, but would like to consider your responses to these concerns and assess a revised manuscript before we make a final decision on publication.

We therefore invite you to revise and resubmit your manuscript, along with a point-by-point response to the reviewers. Please highlight all changes in the manuscript text file.

Editorially, we consider it crucial that the methodological concerns raised by the reviewers, such as the analytic approaches, model complexity, and the assessment of coarticulation, are thoroughly addressed in the revised manuscript. Please also include relevant literature as well as alternative explanations of the central results of the study.

I am attaching an Editorial Requests Table that details critical reporting requirements for the revised manuscript. Please attend to each item and ensure your manuscript is fully compliant. If your revised manuscript is not aligned with these requests on major issues, such as those concerning statistics, it may be returned to you for further revisions without re-review.

Please submit the following items:

- Revised manuscript
- Point-by-point response to the referees' comments
- Cover letter (as a separate document)
- <https://www.nature.com/documents/nr-reporting-summary.pdf> Nature Research Reporting Summary
- Completed Editorial Request Table (attached).

via this link: Link Redacted .

Additional guidance is available in our style and formatting guide Communications Psychology formatting guide.

Best regards,

Troy Lui

Troy Lui, PhD
Associate Editor
Communications Psychology

REVIEWER EXPERTISE:

Reviewer #1: sequential movement/motor control

Reviewer #2: sequential movement/motor control, computational modelling

Reviewer #3: sequential movement/motor control, code review

REVIEWER REPORTS:

Reviewer #1 (Remarks to the Author):

The study presents a novel modelling framework for distinguishing between hierarchical and flat planning of movement sequences (reaching) comprising three-target sequential movements. The authors combine a computational modelling approach with kinematic measurements to assess whether movements are planned as a single, unified trajectory or via a hierarchical control layer that dynamically links simpler movement segments.

The work is timely in light of ongoing debates on sequential motor control and has relevance beyond the motor learning domain. I have several substantive comments relating to the integration with prior literature, the analyses performed, and the clarity of the results.

1. Analyses

My main concern is that the analyses exclude a substantial proportion of the available data. Rather than comparing the full movement trajectories and velocity profiles between empirical data and the two models, the manuscript focuses on a restricted set of kinematic features. The field standard (e.g., Todorov and Jordan, 1998; <https://journals.physiology.org/doi/epdf/10.1152/jn.1998.80.2.696>) is to compare complete trajectories, which would provide a more compelling evaluation of model performance. While the selected features of interest (such as halfway or transitional curvature) are well justified in the context of the research question, the rationale for excluding the remaining data is unclear. I recommend statistically quantifying what is currently shown in Supplementary Figure 3 and integrating this analysis into the main results section. This would strengthen the argument and broaden the conclusions beyond a focus on sub-features of the data.

2. Analyses and visualisation in Figures 4 and following

The simulated results - particularly for the hierarchical model (green circles) - are more widely distributed across the variables of interest than the empirical data. In some sequences (e.g., MoveID4) they diverge substantially from the data, which shows tighter covariance between the plotted features. The metric "percentage of Participant-MoveID combinations covered by each model" excludes model outputs that do not overlap with the empirical data, rather than evaluating the error statistic between model and empirical outputs. This approach risks biasing the metric towards models with maximal variance in their output space, even if their predictions are less specific. As the authors note, HiSeq covers a much larger area than vpSOC(3), but interpret this as a result of increased independence between subsequent actions. Alternative explanations - such as noisier control signals - should be excluded. I would recommended using much more straightforward error-focused statistics such as Euclidian distance to quantify the kinematic output of the different models vs the empirical data.

3. Model visualisation

The modelling framework is described clearly, but a figure illustrating the flat versus hierarchical modules, including

example simulated trajectories, would improve conceptual understanding and provide a compelling visual link between the theoretical framework and the results.

4. Design

It is unclear why target sizes (Targets 1–5) vary. This should be clarified or made more prominent in the Methods section.

5. Literature integration

The introduction is well written and draws on important literature in the field. However, it omits some relevant neuroimaging and behavioural studies on sequence learning and kinematic fusion (e.g., Sporn et al., 2021: <https://journals.physiology.org/doi/full/10.1152/jn.00467.2021>), hierarchical motor sequence planning (Yewbrey et al., 2023: <https://www.jneurosci.org/content/43/10/1742.long>; Yewbrey et al., 2024: <https://www.jneurosci.org/content/44/45/e0832242024>; Ariani et al., 2025: <https://www.jneurosci.org/content/45/3/e1300242024>; Kornysheva et al., 2019: <https://doi.org/10.1016/j.neuron.2019.01.018>) and behavioural evidence for hierarchical motor sequence planning (Mantziara et al., 2021: <https://journals.physiology.org/doi/full/10.1152/jn.00645.2020>). These works provide empirical support for the hierarchical structuring of sequential movement planning and execution, particularly in reaching tasks. Integrating these findings would strengthen the theoretical grounding of the current study which nicely extends the findings applied to kinematically continuous trajectories rather than smooth movements.

In sum, the study offers a novel and relevant approach to modelling sequential movement planning, but its impact would be strengthened by more comprehensive trajectory-based analyses, clearer visualisation of model comparisons, explicit methodological details, and fuller engagement with the existing literature.

Reviewer #2 (Remarks to the Author):

Comments to authors (COMMSPSYCHOL-25-0430-T)

Summary:

Cuevas Rivera and Kiebel investigated how simple sequences of reaching movements, performed without extensive training, are controlled in a hierarchical manner. Specifically, they characterized the kinematic features of sequential reach trajectories executed by human participants using three measures of coarticulation—halfway coarticulation, transitional coarticulation, and trajectory curvature—and compared these with predictions from either a "flat" or a "hierarchical" model of movement control, both based on the via-point stochastic optimal controller (vpSOC). The authors argue that the observed movement sequences reflect hierarchical movement control, as the "hierarchical" vpSOC model accounted for a greater proportion of the variability in the coarticulation measures. The paper is generally well organized and clearly written. However, further discussion may be warranted, as several relevant studies appear to be missing from the citations. When these prior works are considered (as detailed below), the authors' claim that hierarchical movement organization (i.e., chunking) emerges without practice may need to be more cautiously stated. Specific comments follow.

Major Comments:

If I have correctly understood the manuscript, the authors' main claims are:

- (1) that direct behavioral evidence of hierarchical movement organization (i.e., chunking) can be extracted using the sequential reaching paradigm developed in this study, which requires no extensive training; and
- (2) that the behavioral signature of hierarchical movement organization can be tested using their hierarchical sequence model (i.e., vpSOC with stable heteroclinic channels; SHC).

For the reasons outlined below, I suggest the authors consider tempering at least the first claim.

First, several relevant prior studies examining behavioral features of coarticulation or chunking using similar sequential reaching tasks are not cited (e.g., Klein-Breteler et al., *J Neurophysiol*, 2004; Sosnik et al., *Exp Brain Res*, 2004; 2007; Sporn et al., *J Neurophysiol*, 2022). These studies employed comparable methods to extract behavioral indices of coarticulation and often included comparisons with computational models. Notably, some (e.g., Sosnik et al., *Exp Brain Res*, 2004; 2007) have reported that accuracy demands influence the smooth concatenation of movements. I strongly encourage the authors to contextualize their work in light of these previous studies in both the Introduction and Discussion sections, and to clearly articulate how the present study advances beyond these prior findings.

Second, an alternative interpretation should be considered regarding whether the observed coarticulation genuinely reflects a hierarchical organization process (i.e., chunking). Given the task structure—which involves only three targets (thus minimizing memory load) and repeated execution of the same sequence (thus maximizing predictability)—the observed coarticulation could also result from online or pre-planning of multiple movements. These processes do not necessarily imply chunking per se (see, e.g., Ariani et al., *J Neurophysiol*, 2019; 2020; Kashefi et al., *eLife*, 2024). If the authors consider such planning strategies to fall under their definition of hierarchical movement organization, that definition should be clearly and explicitly stated in the Introduction. At present, it appears only later in the Discussion (line 428). In either case, the relationship between the current findings and the aforementioned studies should be addressed directly in the main text.

Comments on the Introduction:

- Lines 26–28:

"Despite the plethora of evidence in neuroimaging studies in both humans and non-human animals, direct behavioral

evidence of the hierarchical organization of movement in the brain has proven difficult to observe (Diedrichsen & Kornysheva, 2015)."

→ The intended meaning here is unclear. Do the authors mean that behavioral evidence is difficult to obtain due to the need for extensive training? The cited review (Diedrichsen & Kornysheva, TiCS, 2015) does not appear to support this specific claim. Please clarify what is meant by "direct behavioral evidence" and in what way it is considered elusive.

- Lines 31–33:

"In this work, we introduce a rapid and training-free approach to obtain immediate and direct behavioral evidence of hierarchical structure in human reaching movements."

→ What exactly is meant by "direct behavioral evidence" in this context? Are the authors implying that previous studies have only reported indirect evidence? If so, this distinction should be justified and clarified.

- Lines 39–40:

"without the confounding effects of extensive training or overlearned behavior"

→ Please explain why and how extensive training or overlearning introduces confounding effects. This point is central to the authors' rationale but is not currently substantiated.

- Line 68:

"First, they open the possibility of directly observing the process of chunking through repetition"

→ As noted above, the observed coarticulation may not necessarily reflect chunking. Furthermore, what constitutes "direct observation" of chunking in this context should be clarified.

- Lines 69–71:

"Secondly, they identify specific moments when higher hierarchical control levels intervene to link different movement segments"

→ Were these "specific moments" empirically identified in the results? If this refers to model simulations, the statement may be too strong and should be rephrased to reflect the nature of the evidence (i.e., that these moments are inferred from simulation, not directly observed).

Model Complexity:

Please report the total number of free parameters for each model. This information is essential to evaluate whether the increased flexibility of the HiSeq model could be explained simply by a greater number of parameters.

Minor Comments:

- There are several minor errors throughout the text and figure captions. For example:

- o The term "hierarchical" is omitted when referring to the HiSeq model (line 104).

- o Degrees of freedom are missing for the reported t-statistics (line 243).

- o Some figure panel labels appear to be incorrect or missing—for instance, the caption for Figure 4 refers to panel IDs that do not exist.

- Optimal feedback control has been conceptualized as comprising multiple components—such as a feedback controller, forward model, and state estimator—each of which has been proposed to map onto distinct brain regions arranged hierarchically (Shadmehr & Krakauer, *Exp Brain Res*, 2008). From this perspective, the vpSOC model could already be considered hierarchical. Could the authors clarify in what specific sense they regard the vpSOC model as not hierarchical?

- Regarding the distinction between the "flat" and hierarchical models: the vpSOC(3) model plans all three movements simultaneously, while the HiSeq model consists of three vpSOC(1) controllers whose outputs are concatenated via SHC dynamics. Conceptually, could these two approaches be considered functionally similar in terms of planning? Clarification on how these models differ with respect to the hierarchical structure of control would be helpful.

Reviewer #3 (Remarks to the Author):

The manuscript addresses an interesting and relevant question in the current landscape of movement neuroscience, that is, how sequences of movements are produced and controlled by the brain. The experiment is overall well designed, but the metrics selected to assess participants' kinematics and evaluate coarticulation lack the conceptual grounding to assess the phenomenon at hand.

1) The key question that we are trying to answer when we assess coarticulation is: given a sequence of movements (e.g., reaching movements like in this case), are the movement parameters extracted from segment n able to predict segment $n+k$? Or, in a more nuanced version, how early the movement parameters extracted from segment n become predictive of segment $n+k$? Note that I am using $n+k$ rather than $n+1$ because coarticulation can span further the next movement and cover a longer chunk of the sequence.

The curvature of the trajectory, or how much the participant cuts the corner at a via point are to an extent epiphenomenal to coarticulation, but per se do not allow to conclude that there is coarticulation.

I encourage the authors to rethink the metrics they adopted by considering this conceptual framework. For example, in MoveID 0, 1, 2, which all start with reaching target 1, is the first segment predictive of the second target? Having answered this participant-wise, they will have a compact measure of coarticulation, that they can now use in their models. The current metrics make the results very hard to interpret in terms of coarticulation.

2) Considering the above, it is unclear to me how half-way and transitional coarticulation are different “types” of coarticulation. What distinct aspects of the process of linking two subsequent movement segments do these metrics reflect? I am not convinced this comes across clearly in the manuscript, which makes the use of these two measures of coarticulation not fully justified.

3) In terms of writing, I would also encourage the authors to reduce the amount of jargon and acronyms in the manuscript – as it is, the text is not easy to follow, and one needs to keep going through the glossary. For example, why call the sequences MoveID, which is very peculiar, and not just sequence 1, 2, etc? More in general, I suggest revising the manuscript in a way that reduces to a minimum the very need for a glossary.

4) Some figures are also hard to read. For example, Figure 2 is clear with respect to the trajectories, whereas plotting velocities from all trials on top of each other makes individual traces very difficult to appreciate. I encourage the authors to plot the average velocity accompanied (if needed) by individual trials representative of features of interest they wish to highlight for the reader. Also related to visualisation, in Figure 3, the x axis reflects the sequence, which is now called Sq0, 1, 2 etc, instead of MoveID as before. I strongly suggest adopting common naming conventions.

The code seems functional and well-documented. Also the adopted analysis are described in detail in the paper. As I discuss in my comments, my concerns are mostly at conceptual/scientific level.

Version 1:

Decision Letter:

Dear Dr Cuevas Rivera,

Your manuscript titled "Behavioral evidence for the hierarchical execution of sequential movements" has now been seen by our reviewers, whose comments appear below. In light of their advice I am delighted to say that we are happy, in principle, to publish a suitably revised version in Communications Psychology.

We therefore invite you to revise your paper one last time to address the remaining concerns of our reviewers and a list of editorial requests. At the same time we ask that you edit your manuscript to comply with our format requirements and to maximise the accessibility and therefore the impact of your work.

EDITORIAL REQUESTS:

The reviewers highlight a lack of clarity on how the work adds to the existing corpus of research. Please delineate more clearly how the work confirms, extends, or contradicts previous studies, avoiding overt novelty claims.

SUBMISSION INFORMATION:

OPEN ACCESS:

* DATA AVAILABILITY:

Link Redacted

Best regards,

Troy Lui

Troy Lui, PhD
Associate Editor
Communications Psychology

REVIEWERS' COMMENTS:

Reviewer #1 (Remarks to the Author):

The authors have provided a thorough and thoughtful response to all my queries, and have strengthened their paper by improving its integration with previous work. I do not have any further substantial queries.

Reviewer #2 (Remarks to the Author):

The authors have addressed most of my previous comments, and the manuscript has become substantially clearer. However, the argument presented in lines 32–49 is still not sufficiently developed to clearly distinguish the current study

from previous work. Further elaboration on this point would help sharpen the contrast with earlier studies and strengthen the overall contribution of the paper.

Specific Comments (lines 32-49):

“Despite the plethora of evidence in neuroimaging studies in both humans and non-human animals, direct behavioral evidence of the hierarchical organization of movement in the brain has proven difficult to observe (Diedrichsen & Kornysheva, 2015). Instead, behavioral studies have focused on measures related to chunking (e.g. Sakai et al., 2003; Sporn et al., 2022; Wymbs, Basset, Mucha, Porter, & Grafton, 2012) and to pre-planning (Ariani & Diedrichsen, 2019; Mantziara et al., 2021; Verwey, Abrahamse, Ruitenberg, Jiménez, & de Kleine, 2011). However, how these measures are related to the hierarchical organization of movements is still unclear. Behavioral studies typically rely on long training sessions (Verwey et al., 2011), often across multiple days to induce chunking (Sakai et al., 2003; Wymbs et al., 2012). Furthermore, these studies have used extensive training to observe their results, which has been shown to reduce reaction times, error rates, inter-movement intervals, and movement fluidity even for untrained sequences (Acuna et al., 2014; Ariani & Diedrichsen, 2019; Kornysheva et al., 2019; Yewbrey, Mantziara, & Kornysheva, 2023), affecting results based on these measures. In this work, we introduce a rapid and training-free approach to obtain immediate and direct behavioral evidence of hierarchical structure in human reaching movements. Importantly, we show that even without extensive training, there is readily accessible and clear evidence in behavior that many sequential movements are organized hierarchically.”

Here, the authors state “However, how these measures are related to the hierarchical organization of movements is still unclear.” but providing one or two concrete examples would help clarify and strengthen this point. Although the authors mentioned “Furthermore, these studies have used extensive training to observe their results, which has been shown to reduce reaction times, error rates, inter-movement intervals, and movement fluidity even for untrained sequences [...], affecting results based on these measures.” later in the text, it would be helpful to explain more explicitly why this is considered problematic. For instance, is the concern that behavioral measures of hierarchical structure in the trained sequences might be obscured by an apparent improvement in an untrained sequences, or is a more fundamental issue being suggested? Clarifying this would improve the reader’s understanding of the authors’ argument.

Also, after reading the manuscript several times, the claim of “direct evidence” of hierarchical structure still appears somewhat strong. In the current form, the hierarchical structure seems to be inferred from a fixed set of computational models, with model selection based on comparing the range of predicted kinematic indices across models against the observed data. This approach is informative but remains relatively indirect. While the comparison may rule out flat model (vpSOC), it does not necessarily establish that the alternative hierarchical model (HiSeq) is the correct account. To strengthen this point, the authors may wish to either temper the wording of the “direct evidence” claim or provide additional analyses or data. For example, fitting the free parameters of the HiSeq model and examining whether a single parameter set (or at least a constrained subset of parameters) can reliably reproduce the observed data would provide stronger support for the proposed interpretation. In addition, the authors may consider whether the model yields testable predictions that could be examined in an independent experiment (e.g., changing target sizes and arrangements). Demonstrating that a novel prediction derived from the model is fulfilled would provide particularly strong support for the claim of hierarchical structure without additional training and would substantially strengthen the interpretation as “direct evidence.” Even if such an experiment is beyond the scope of the current study, clarifying these predictions would help delineate the model’s explanatory power.

Reviewer #3 (Remarks to the Author):

I appreciate that the authors have addressed my comments in the revised version of the manuscript. The text has greatly improved from the previous version and is now clearer and easier to read.

Yet, a few concerns remain, especially with respect to the novelty of the results presented here and the framing with the current literature. In the new version of the introduction, the authors claim that “how these measures [e.g., chunking and anticipatory planning] are related to the hierarchical organization of movements is still unclear”. In what way it is not clear? To me, chunking looks very much like a proxy of hierarchical movement control. In other words, it is not clear how the model-based approach proposed here unveils the hierarchical structure of voluntary movement more effectively than chunking.

Subsequently, the authors argue that “these [previous] studies have used extensive training to observe their results, which has been shown to reduce reaction times, error rates, inter-movement intervals, and movement fluidity even for untrained sequences, affecting results based on these measures.” But still, in spite of these issues, previous work has nevertheless provided evidence of the hierarchical organisation of voluntary movement. I can see the practical advantage of a paradigm that requires shorter or no training, but what new discovery about how the brain controls movements does this new approach foster?

I believe the work could further benefit from clarifying the answer to this question throughout the manuscript – that is, to what extent these results offer an elegant confirmation of what is already largely known about the hierarchical control of movement using a more efficient task vs. how much these results constitute instead a new discovery.

Response to reviewers

Dario Cuevas Rivera, Stefan Kiebel

November 25, 2025

We would like to thank all three reviewers for their thoughtful and constructive feedback, which has greatly improved the clarity and focus of our manuscript. Below, we address each comment in turn, organized by reviewer. Text citations from the manuscript are indented, and new or revised text is shown in red. In addition to these changes, we made small changes to the manuscript to improve style and legibility. These minor changes are highlighted in the manuscript (with change tracking), but not explicitly listed here.

Contents

1	Reviewer 1	1
1.1	Analyses	2
1.2	Analyses and visualisation in Figures 4 and following	3
1.3	Model visualisation	5
1.4	Design	5
1.5	Literature integration	5
1.6	Summary	6
2	Reviewer 2	6
2.1	Major Comments	7
2.2	First comment	7
2.3	Second comment	8
2.4	Comments on the Introduction	8
2.4.1	Lines 26-28	8
2.4.2	Lines 31-33	9
2.4.3	Lines 39-40	9
2.4.4	Line 68	10
2.4.5	Lines 69-71	10
2.4.6	Model Complexity	10
2.5	Minor Comments	11
2.6	Optimal feedback control	11
2.7	Distinction between flat and hierarchical	12
3	Reviewer 3	12
3.1	Comment 1	12
3.1.1	Curvature and coarticulation	13
3.2	Comment 2	14
3.3	Comment 3	14
3.4	Comment 4	15
4	References	15

1 Reviewer 1

The study presents a novel modelling framework for distinguishing between hierarchical and flat planning of movement sequences (reaching) comprising three-target sequential movements. The authors combine a computational modelling approach with kinematic measurements to assess whether movements are planned as a single, unified trajectory or via a hierarchical control layer that dynamically links simpler movement segments.

The work is timely in light of ongoing debates on sequential motor control and has relevance beyond the motor learning domain. I have several substantive comments relating to the integration with prior literature, the analyses performed, and the clarity of the results.

1.1 Analyses

My main concern is that the analyses exclude a substantial proportion of the available data. Rather than comparing the full movement trajectories and velocity profiles between empirical data and the two models, the manuscript focuses on a restricted set of kinematic features. The field standard (e.g., Todorov and Jordan, 1998) is to compare complete trajectories, which would provide a more compelling evaluation of model performance. While the selected features of interest (such as halfway or transitional curvature) are well justified in the context of the research question, the rationale for excluding the remaining data is unclear. I recommend statistically quantifying what is currently shown in Supplementary Figure 3 and integrating this analysis into the main results section. This would strengthen the argument and broaden the conclusions beyond a focus on sub-features of the data.

We thank the reviewer for this suggestion. Before starting our data analyses, we considered performing full Bayesian model fitting with MCMC methods to analyze full trajectories and compare models, as we have done in recent works (Cuevas Rivera et al., 2020; Legler et al., 2025). However, we decided against this approach because of the disadvantages inherent to model fitting which would also translate to fitting trajectories. Instead, we followed the approach used before in motor control (e.g. Guigon et al., 2019; Yeo et al., 2016; Crevecoeur et al., 2020), relying on relevant features of the movements (Wymbs et al., 2012; Sporn et al., 2022), instead of the movements themselves, to compare model capabilities to observed participant behavior.

In the following, we will first provide a detailed explanation of the underlying rationale to focus on specific features instead of full trajectory analysis. We then cite the text that we added to the "Classifying trajectories as flat or hierarchical" section in Results.

Rationale for focusing on features and forgoing of full trajectory analysis

As the reviewer pointed out, we decided to focus on features of the curves produced by participants that have been recently shown to be relevant to sequential movements (e.g. Kashefi et al., 2024), specifically trajectory curvature and coarticulation. This is in line with recent studies, where these and similar measures have been used to study chunking and, in general, the effects of practice in sequential movements (Sakai et al., 2003; Wymbs et al., 2012; Sporn et al., 2022). Our approach has the added advantage of a focus on whether a model is even capable of reproducing an observed measure or not, sidestepping the sliding scale of an error-based model fitting procedure.

The most important disadvantage from an approach based on model fitting is related to the interpretability of the results. To show this, we fitted both models to our participant data and show the resulting best-fitting trajectories in figure 1 below. The fitting procedure is described in the figure caption. In the selected example, the error between observed data and best-fitting curves is close between the models, with a slight advantage for HiSeq. With this we can highlight three main problems with this approach: (1) Qualitatively, the fit differs between models in ways that are difficult to interpret. Any ad-hoc descriptions of the differences (e.g., that HiSeq overshoots the line that connects the first target to the second) more than vpSOC(3) we found to be not consistent across participant-sequence combinations. (2) The error measure used during model fitting (DTW-corrected Euclidean distance) will sacrifice model fit in relevant measures (such as coarticulation and curvature) in favor of irrelevant features, such as noise in the trajectories, small variability in the timing, and artifacts created by pre-processing of the data (such as the DTW correction applied in this case). While some of these can be ameliorated, their effects are difficult to estimate. (3) In this example, using Euclidean distance leaves us with the question of significance. How big a difference in distances is enough for a model to be significantly better than the other? Guidelines exist for the interpretation of model comparison tools (e.g. Gelman et al., 2014). However, in our case, we

Figure 1: Model fitting for one participant, one sequence, averaged across all repetitions. The experimentally observed trajectory is plotted in black, the HiSeq fit in blue and the vpSOC(3) fit in red. In the legend, the sum of all Euclidean distances between each model and the experimental data is shown, in mm. To obtain this measure, trajectories were cut to start after leaving the initial circle, to before re-entering, to equalize movement durations. Trajectories were then resampled to 100 points, equally spaced (as measured with arc length) across each trajectory. The overall distance is the average across all 100 points.

found model fits to be similar for most trajectories, in part due to the flexibility of the models, and in part due to the error measure itself, which can be minimized with trajectories that are, in terms of motor control-relevant features, very different from each other.

Text added to the Results section (with surrounding text included for clarity; new text highlighted)

We now build on the correlations from the previous section to categorize participants’ movements as being consistent with either flat or hierarchical planning. To do this, we analyze the maximum curvature of trajectories (see Methods) as they transition from one target to the next, a quantity related to transitional coarticulation. We will show that HiSeq’s disentanglement of halfway and transitional coarticulation matches what we observed in participant data for many trajectories. Using these measures, we specify two criteria to determine whether a participant trajectory is compatible only with hierarchical planning (HiSeq), only with non-hierarchical planning (vpSOC(3)), or with both. **This approach has two advantages: (1) it specifically relies on features that have been shown recently to be related to sequential movements and chunking (e.g. Kashefi et al., 2024); (2) it directly tests whether each model can reproduce the observed combinations of coarticulation and curvature, establishing absolute limitations of their explanatory power.**

1.2 Analyses and visualisation in Figures 4 and following

The simulated results - particularly for the hierarchical model (green circles) - are more widely distributed across the variables of interest than the empirical data. In some sequences (e.g., Sequence 4) they diverge substantially from the data, which shows tighter covariance between the plotted features. The metric “percentage of Participant–MoveID combinations covered by each model” excludes model outputs that do not overlap with the empirical data, rather than evaluating the error statistic between model and empirical outputs. This approach risks biasing the metric towards models with maximal variance in their output space, even if their predictions are less specific. As the authors note, HiSeq covers a much larger area than vpSOC(3), but interpret this as a result of increased independence between subsequent actions. Alternative explanations - such as noisier control signals - should be excluded. I would recommended using much more straightforward error-focussed statistics such as Euclidian distance to quantify the kinematic output of the different models vs the empirical data.

Yes, we agree with the reviewer that our analyses favor flexible models. This reflects our interpretation that the flat model (vpSOC(3)) lacks sufficient flexibility to account for the observed variability in the data. Our primary focus of the analysis was on model exclusion, with which we showed that some trajectories cannot possibly be modeled with vpSOC(3). Regarding motor noise in control signals, we controlled for this by considering only simulations without noise during execution (although the models were initialized with noise), and we account for noise in the observed trajectories by considering the 95% confidence interval, and determining whether it was included in the area of coverage of the models.

To clarify this, we have made several changes to the manuscript:

1. We have added the number of model parameters to the Methods section, to enable a discussion about model complexity. Concretely:

The number of parameters for vpSOC(3) is 17, and for HiSeq 20. However, most of these parameters are determined by the task, or were fixed identically across both models in all simulations. This includes the three parameters unique to HiSeq (*monitor_distance*, *monitor_sd* and τ_{23}), which were set to match the target timing observed in participant trajectories. Note that vpSOC(N) can in principle include separate timing parameters for each segment, adding up to N free parameters per sequence. In our simulations, however, we fixed these timings as equally spread (just as we fixed the timing-related parameters in HiSeq), so we did not treat them as free parameters in vpSOC(3).

What matters more than the total number of parameters of each model, is how many parameters we varied to explain the behavioral data. The explanatory power of each model, as seen in Figure 4 and Figure 5, was obtained by changing 3 parameters for both models: r , h_1 and h_2 .

2. To address the reviewer’s comment that HiSeq covers areas beyond the experimental trajectories, we have added the following text in the section ”Classifying trajectories as flat or hierarchical” of the Results:

As can be clearly seen in Figure 4, HiSeq covers a much wider area than vpSOC(3), including more of our participant data: For instance, multiple participant-sequence combinations show a much higher curvature for highly-negatively coarticulated trajectories (i.e. those between x-values of -20 to -10 within the green area but not the black area). These trajectories display a flexibility that cannot be explained with vpSOC(3), suggesting that can only be explained by the hierarchical structure of HiSeq. Such an analysis can be seen as biased in favor of a more complex model, but this lack of complexity in vpSOC(3) is central to our reasoning, suggesting that the brain uses additional tools for sequential motor planning, than used for the planning of simpler, single-target movements.

3. Regarding motor noise as a possible confound, we have added the following clarification:

We quantified how well each model explains the observed trajectories by assessing the proportion of trials covered by their respective prediction spaces in Figure 4 and Figure 5. To account for the effects of motor and observation noise in the observed trajectories, we combined the area-of-coverage comparison with a significance test to conclude that some trajectories cannot possibly be modeled with vpSOC(3). To do this, we calculated the 99% confidence intervals for the mean of each star (see Methods), and determined whether it intersected each model’s coverage. The results are summarized in Table 1, which includes not only the classification criteria from Figure 4 and Figure 5, but also coverage based on the maximum (negative) hC^1 . Overall, neither model alone explains the majority of participant behavior: HiSeq accounts for up to 49%, and vpSOC(3) only 9%. However, a version of HiSeq that incorporates both vpSOC(2)+vpSOC(1) and vpSOC(3) trajectories would account for 69% of the observed variability. These results that the flat planning approach of vpSOC(N) is limited in its scope, and suggests tha the observed trajectories were produced with a hierarchical approach to motor planning.

4. To address model complexity and number of parameters, we added the following text at the end of the section ”Alternative formulations” in the Discussion:

As shown in Figure 4 and Figure 5, HiSeq covers an area in the curvature and halfway coarticulation spaces that is wider than that observed in our experimental data. This is due partially to our choice of plotting all simulated trajectories for all model parameter values in every panel for completeness. In addition, HiSeq and, to a lesser extent vpSOC(N), can generate trajectories that were not observed in our study. With our sample of trajectories from 20 participants, we cannot determine whether these unobserved trajectories are also part of the broader human repertoire. In principle, a Bayesian approach could be used in a future study to limit these models' flexibility to those observed in a significant sample of the general population: by using Bayesian model fitting techniques, a larger data set could be used to obtain posteriors over parameters, which could subsequently be used as priors for the entire population.

5. We have also added a related discussion to the "Limitations and improvements" section in the Discussion:

Previous approaches have shown that even simpler models than SOC can be fitted very closely to hand-movement trajectories during sequential reaching tasks (e.g. Todorov and Jordan, 1998; Sosnik et al., 2004, 2007). In this work, we chose not to perform model fitting, instead focusing on a method by exclusion, with which we showed that some trajectories cannot possibly be modeled with vpSOC(3) (see Table 1). Model fitting would not necessarily provide further insights in this regard, but it could have important uses in future works. For instance, an important advantage of model fitting is that it enables the use of the fitted models for prediction of future movements. Furthermore, it would enable the search for neural correlates of the model variables in neuroimaging studies (Fine et al., 2017; Yokoi and Diedrichsen, 2019).

1.3 Model visualisation

The modelling framework is described clearly, but a figure illustrating the flat versus hierarchical models, including example simulated trajectories, would improve conceptual understanding and provide a compelling visual link between the theoretical framework and the results.

We thank the reviewer for this suggestion, and now include a new figure in the methods section (new figure 6), which contains a diagram of the HiSeq model. Additionally, we have moved some of the simulations from the Supplementary Materials to this figure, to exemplify the trajectories produced by both models.

The parametrization for these simulations has been additionally added to the configuration file `paper.conf.py` included in the submission.

1.4 Design

It is unclear why target sizes (Targets 1–5) vary. This should be clarified or made more prominent in the Methods section.

We apologize for this omission. We have added the following explanations:

1. A quick note in the Experiment section of the Results:

Participants (N=20) sat at a table with a printout on it which displayed large colored circles that were to be the targets of reaching movements. The printout can be seen in Figure 1A, where the small gray circle (7 cm in diameter) labeled with 0 is the starting position of all movements, and the other five colored circles are potential targets. These targets were divided into primary (1: blue and 2: red; 11 cm in diameter) and secondary (3: olive, 4: lilac, 5: pink; 7 cm, 9 cm and 7 cm in diameter, respectively). **The different sizes of the targets were chosen to foster coarticulation, especially around the primary targets, while allowing for straighter movements towards the top secondary target.** See Methods for the exact dimensions and distances of the printout, as well as the specification of all parameters of the experiment.

2. The full explanation in the Methods:

The sizes were chosen as follows: (1) The primary targets (blue and red) were chosen to be the largest to lower demands for accuracy. Given the results from previous experiments (e.g. Rand et al., 1997; Rand and Stelmach, 2000), we expected this reduction in accuracy to result in higher coarticulation. The secondary targets were reduced in size to be able to fit the entire printout in a comfortable workspace in front of the participants, while maintaining separation between the different targets. The exception to this is the top secondary target, which is larger than the ones on the sides. The size of this top target was chosen to enable participants to complete the upward trajectory in a straight line, for trajectories including this secondary target (sequences 2 and 5). For example, in sequence 2 (start-blue-lilac-start), a straight-line trajectory could touch the right edge of the blue target and the left edge of the lilac target, then move back.

1.5 Literature integration

The introduction is well written and draws on important literature in the field. However, it omits some relevant neuroimaging and behavioural studies on sequence learning and kinematic fusion (e.g., Sporn et al., 2021: <https://journals.physiology.org/doi/full/10.1152/jn.00467.2021>), hierarchical motor sequence planning (Yewbrey et al., 2023; Yewbrey et al., 2024; Ariani et al., 2025; Kornysheva et al., 2019) and behavioural evidence for hierarchical motor sequence planning (Mantziara et al., 2021:). These works provide empirical support for the hierarchical structuring of sequential movement planning and execution, particularly in reaching tasks. Integrating these findings would strengthen the theoretical grounding of the current study which nicely extends the findings applied to kinematically continuous trajectories rather than smooth movements.

We thank the reviewer for the suggested literature and have added discussions to relate our results to the now cited papers in the Introduction.

Of particular note is Sporn (2021), which we found strongly relates to our results and supports our inference. We have now incorporated their fusion index analysis into our results section to show that our classification of trajectories into hierarchical and flat is related to the fusion index. In particular, we show that for HiSeq-like trajectories, the fusion index, as expected, is significantly lower around the second target, i.e., the target around which the vpSOC(2) segment is concatenated with the following vpSOC(1), than for vpSOC(3)-like trajectories. These new results are shown in the new figure 6 and surrounding text, which is as follows:

Given the relation between the hierarchical organization of movements with chunking, an important question is if our model-based classification is related to existing measures of chunking. Recently, the fusion index was proposed as a measure of chunking (Sporn et al., 2022). The fusion index reflects the difference between the minimum speed around an intermediary target and the maximum speeds of the segments that surround it (see Fusion index and bootstrap section in Methods for the mathematical definition). The fusion index is in the interval $[0, 1]$; a value of 1 means the trajectories are fused, i.e. all targets are reached in one swipe. A value of 0 means the trajectories are performed in segments.

We first show in Figure 6A that halfway coarticulation hC^1 is highly correlated to the fusion index, similar to the findings by (Sporn et al., 2022). This correlation does not hold for trajectories with positive hC^1 , which is likely due to the geometry in the task, which does not allow very high values of positive hC^1 , as well as the low number of positively-coarticulated trajectories for some sequences, with sequence 2 having none.

We expected our classification of trajectories to lead to vpSOC(3)-like trajectories having a higher fusion index (Sporn et al., 2022) than HiSeq-like trajectories. In Figure 6B we show the distribution of fusion index values for both models, separated by segment. The first segment ends around the first target, and the second segment around the second target. As expected, the mean fusion index (green triangles) across trajectories for both segments are higher for vpSOC(3) than for HiSeq. Importantly, given that HiSeq uses one vpSOC(2) for the first two targets, the fusion index is also lower for the second segment than for the first. To rule out the possibility that these differences between groups were due to random chance, we performed a bootstrap analysis (see Fusion index and bootstrap section in Methods for details) and found a probability $p < 0.01$ of finding a difference between groups as large or larger than that of our model-based classification.

1.6 Summary

In sum, the study offers a novel and relevant approach to modelling sequential movement planning, but its impact would be strengthened by more comprehensive trajectory-based analyses, clearer visualisation of model comparisons, explicit methodological details, and fuller engagement with the existing literature.

2 Reviewer 2

Cuevas Rivera and Kiebel investigated how simple sequences of reaching movements, performed without extensive training, are controlled in a hierarchical manner. Specifically, they characterized the kinematic features of sequential reach trajectories executed by human participants using three measures of coarticulation—halfway coarticulation, transitional coarticulation, and trajectory curvature—and compared these with predictions from either a "flat" or a "hierarchical" model of movement control, both based on the via-point stochastic optimal controller (vpSOC). The authors argue that the observed movement sequences reflect hierarchical movement control, as the "hierarchical" vpSOC model accounted for a greater proportion of the variability in the coarticulation measures. The paper is generally well organized and clearly written. However, further discussion may be warranted, as several relevant studies appear to be missing from the citations. When these prior works are considered (as detailed below), the authors' claim that hierarchical movement organization (i.e., chunking) emerges without practice may need to be more cautiously stated. Specific comments follow.

2.1 Major Comments

If I have correctly understood the manuscript, the authors' main claims are: (1) that direct behavioral evidence of hierarchical movement organization (i.e., chunking) can be extracted using the sequential reaching paradigm developed in this study, which requires no extensive training; and (2) that the behavioral signature of hierarchical movement organization can be tested using their hierarchical sequence model (i.e., vpSOC with stable heteroclinic channels; SHC). For the reasons outlined below, I suggest the authors consider tempering at least the first claim.

2.2 First comment

*First, several relevant prior studies examining behavioral features of coarticulation or chunking using similar sequential reaching tasks are not cited (e.g., Klein-Breteler et al., *J Neurophysiol*, 2004; Sossnik et al., *Exp Brain Res*, 2004; 2007; Sporn et al., *J Neurophysiol*, 2022). These studies employed comparable methods to extract behavioral indices of coarticulation and often included comparisons with computational models. Notably, some (e.g., Sossnik et al., *Exp Brain Res*, 2004; 2007) have reported that accuracy demands influence the smooth concatenation of movements. I strongly encourage the authors to contextualize their work in light of these previous studies in both the Introduction and Discussion sections, and to clearly articulate how the present study advances beyond these prior findings.*

We thank the reviewer for these references. We have followed the reviewer's suggestion and both added these references to the manuscript (Results and Discussion) and now discuss our results in the context of the provided references. We have further added how our work advances on these previous results. In particular, we have added the following texts:

1. We have added the provided references to the Introduction, discussing experimental designs and how our experiment differs from these previous works. Concretely, we added the following:
Crucially, similarly to previous studies (Sossnik et al., 2004, 2007; Kashefi et al., 2024), coarticulation in our task happens at the kinematic level, specifically in the trajectory of the fingertip as it goes from one target circle to the next. Contrary to previous experiments (Sossnik et al., 2004, 2007; Rand et al., 1997; Rand and Stelmach, 2000; Kashefi et al., 2024), we designed an experiment with large targets to lower accuracy demands and foster coarticulation.

2. We have added a discussion about model fitting, connecting it to the work in Sosnik et al. (2004, 2007). Concretely, we have added the following text:

As shown in Figure 4 and Figure 5, HiSeq covers an area in the curvature and halfway coarticulation spaces that is wider than that observed in our experimental data. This is due partially to our choice of plotting all simulated trajectories for all model parameter values in every panel for completeness. In addition, HiSeq and, to a lesser extent vpSOC(N), can generate trajectories that were not observed in our study. With our sample of trajectories from 20 participants, we cannot determine whether these unobserved trajectories are also part of the broader human repertoire. In principle, a Bayesian approach could be used in a future study to limit these models' flexibility to those observed in a significant sample of the general population: by using Bayesian model fitting techniques, a larger data set could be used to obtain posteriors over parameters, which could subsequently be used as priors for the entire population.

3. Of particular note is Sporn et al. (2022): related to the following reviewer's comment, we have added calculations of the fusion index to the Results section in the new figure 6 to show that vpSOC(3)-like trajectories have a higher fusion index than HiSeq-like trajectories, thereby connecting our results to the measures of chunking from Sporn et al. (2022). The following text was added alongside the new figure 6:

We expected our classification of trajectories to lead to vpSOC(3)-like trajectories having a higher fusion index than HiSeq-like trajectories. In Figure 6B we show the distribution of fusion index values for both models, separated by segment. The first segment ends around the first target, and the second segment around the second target. As expected, the mean fusion index (green triangles) across trajectories for both segments are higher for vpSOC(3) than for HiSeq. Importantly, given that HiSeq uses one vpSOC(2) for the first two targets, the fusion index is also lower for the second segment than for the first. To rule out the possibility that these differences between groups were not due to random chance, we performed a bootstrap analysis (see Methods for details) and found a probability $p < 0.01$ of finding a difference between groups as large or larger than that of our model-based classification.

2.3 Second comment

Second, an alternative interpretation should be considered regarding whether the observed coarticulation genuinely reflects a hierarchical organization process (i.e., chunking). Given the task structure—which involves only three targets (thus minimizing memory load) and repeated execution of the same sequence (thus maximizing predictability)—the observed coarticulation could also result from online or pre-planning of multiple movements. These processes do not necessarily imply chunking per se (see, e.g., Ariani et al., J Neurophysiol, 2019; 2020; Kashefi et al., eLife, 2024). If the authors consider such planning strategies to fall under their definition of hierarchical movement organization, that definition should be clearly and explicitly stated in the Introduction. At present, it appears only later in the Discussion (line 428). In either case, the relationship between the current findings and the aforementioned studies should be addressed directly in the main text.

This is indeed a crucial point that we had not discussed in the manuscript. We have now added (1) a clarification on the relation between chunking and our models, and (2) a discussion on how the planning performed by the two models relates to recent experimental results. Concretely, we have added the following clarification on planning strategies to the Introduction (new text in red):

We used two quantitative measures of coarticulation in combination with computational models to differentiate between flat (i.e. non-hierarchical) and hierarchical planning. To do this, we model the two planning strategies with two closely matched models. Both models are grounded in the same optimization framework, the stochastic optimal controller (SOC) (Todorov, 2005), and use identical movement primitives based on via-point control. The key difference lies in their architecture: in the first, a flat model plans the full sequence as a single, unified trajectory. In the second, a hierarchical model introduces a minimal and neurobiologically plausible control layer that dynamically links simpler movement segments. This extension allows the hierarchical model to flexibly compose multi-target sequences in real time without altering the underlying control principles. **Importantly, both models can plan movements in their entirety and differ only in which elements are required before movement onset: vpSOC requires all targets and control parameters to be planned before movement onset, consistent with the concept of chunking (Ramkumar et al., 2016) and competitive queuing (e.g. Yewbrey et al., 2023), while HiSeq can start by planning the first target(s) and add more as needed, consistent with recent observations regarding planning horizons in a similar sequential task (Kashefi et al., 2024).** By holding the core control principles and simple movements constant across models, we ensure that observed differences in fit arise from the presence of hierarchical structure, and not simply from a richer set of low-level parameters.

2.4 Comments on the Introduction

2.4.1 Lines 26-28

“Despite the plethora of evidence in neuroimaging studies in both humans and non-human animals, direct behavioral evidence of the hierarchical organization of movement in the brain has proven difficult to observe (Diedrichsen & Kornysheva, 2015).” → *The intended meaning here is unclear. Do the authors mean that behavioral evidence is difficult to obtain due to the need for extensive training? The cited review (Diedrichsen & Kornysheva, TiCS, 2015) does not appear to support this specific claim. Please clarify what is meant by “direct behavioral evidence” and in what way it is considered elusive.*

We apologize for the ambiguity in the statement. We now clarify what we meant by hierarchical organization. To do this, we have added the following text (highlighted in red) to the manuscript:

1. In the Introduction, we now specify that we mean a hierarchy of controllers, i.e. that a simple controller might produce motor commands, while a higher-level controller might deploy simpler controllers to create sequences. The added text is as follows:

Body movements are known to be hierarchically **controlled** in the brain. Evidence for such a hierarchy **of controllers** has been found in neuroimaging and lesion studies of animals and humans, where simple, often short and single-target movements have been found to be learned and executed within the primary motor cortex (Yokoi and Diedrichsen, 2019; Kawai et al., 2015; Overduin et al., 2012), while the execution of complex movements has been found in pre-motor and parietal areas (Yokoi et al., 2018; Yokoi and Diedrichsen, 2019), and learning in basal ganglia (Tremblay et al., 2009) and the supplementary motor area (Verwey et al., 2002). **In this hierarchy, the simple controllers in the primary motor cortex could be deployed by pre-motor and parietal areas to create sequences of these single-target movements.**

2. Regarding the direct behavioral evidence, the intended meaning was that the studies that have shown behavioral evidence for this hierarchy have focused on error rates, reaction times and evidence for chunking. While these measures, especially chunking, could be potentially established to be related to the hierarchy of controllers we discuss in our work, we focused instead on a more direct approach, using the two proposed models to infer the hierarchy. Besides being a more direct approach, this had two advantages: (1) it did not require long training sessions to compare movements before and after; (2) it can accommodate partially-chunked trajectories, where e.g. the first two targets are chunked, but the third is not, such as those found recently by Kashefi et al. (2024). To clarify this, we have modified the following paragraph in the introduction (with new text highlighted here):

Despite the plethora of evidence in neuroimaging studies in both humans and non-human animals, direct behavioral evidence of the hierarchical organization of movement in the brain has proven difficult to observe. **Instead, behavioral studies have focused on measures related to chunking (e.g. Sakai et al., 2003; Wymbs et al., 2012; Sporn et al., 2022) and to pre-planning (e.g. Verwey et al., 2011; Mantziara et al., 2021; Ariani and Diedrichsen, 2019). However, how these measures are related to the hierarchical organization of movements is still unclear.**

2.4.2 Lines 31–33

“In this work, we introduce a rapid and training-free approach to obtain immediate and direct behavioral evidence of hierarchical structure in human reaching movements.” → What exactly is meant by “direct behavioral evidence” in this context? Are the authors implying that previous studies have only reported indirect evidence? If so, this distinction should be justified and clarified.

In response to the previous comment, we modified the sentence leading up to the one quoted here by the reviewer. In the new text, we address how we consider the evidence of previous studies to be an indirect measure of the hierarchy of controllers. The modified text is as follows:

Despite the plethora of evidence in neuroimaging studies in both humans and non-human animals, direct behavioral evidence of the hierarchical organization of movement in the brain has proven difficult to observe. **Instead, behavioral studies have focused on measures related to chunking (e.g. Sakai et al., 2003; Wymbs et al., 2012; Sporn et al., 2022) and to pre-planning (e.g. Verwey et al., 2011; Mantziara et al., 2021; Ariani and Diedrichsen, 2019). However, how these measures are related to the hierarchical organization of movements is still unclear.**

2.4.3 Lines 39–40

“without the confounding effects of extensive training or overlearned behavior” → Please explain why and how extensive training or overlearning introduces confounding effects. This point is central to the authors’ rationale but is not currently substantiated.

We apologize for this omission. We have added the following text to the Introduction:

Instead, behavioral studies have focused on measures related to chunking (e.g. Sakai et al., 2003; Wymbs et al., 2012; Sporn et al., 2022) and to pre-planning (Verwey et al., 2011; Mantziara et al., 2021; Ariani and Diedrichsen, 2019). **However, how these measures are related to the hierarchical organization of movements is still unclear. Furthermore, these studies have used extensive training to observe their results, which has been shown to reduce reaction times, error rates, inter-movement intervals, and movement fluidity even for untrained sequences (e.g. Acuna et al., 2014; Ariani and Diedrichsen, 2019; Kornysheva et al., 2019; Sosnik et al., 2007; Yewbrey et al., 2023), affecting results based on these measures.**

2.4.4 Line 68

“First, they open the possibility of directly observing the process of chunking through repetition” → As noted above, the observed coarticulation may not necessarily reflect chunking. Furthermore, what constitutes “direct observation” of chunking in this context should be clarified.

We agree with the reviewer in this and the next point. As a consequence, we have rewritten the introduction’s last paragraph in terms of the discussions above regarding hierarchy, chunking and what we mean by direct behavioral evidence. The paragraph now reads:

Our methodology and results have three important implications for further studies. First, they allow detecting the differences between hierarchical and flat movement planning strategies, without relying on measures that require comparisons across hundreds of trials of training that might affect even untrained movements. Second, they allow for this classification even in tasks with continuous hand movements where error rates are practically zero, as is the case in tasks with low accuracy constraints. Third, by connecting flat planning to chunking via the fusion index, our results could help establish the mechanisms behind the process of chunking. Together, these advantages enable the study of chunking in motor tasks in which participants can be highly proficient and produce smooth, highly-coarticulated trajectories.

2.4.5 Lines 69–71

“Secondly, they identify specific moments when higher hierarchical control levels intervene to link different movement segments” → Were these “specific moments” empirically identified in the results? If this refers to model simulations, the statement may be too strong and should be rephrased to reflect the nature of the evidence (i.e., that these moments are inferred from simulation, not directly observed).

We agree with the reviewer that this statement was too strong and have removed it from the introduction. Instead, we briefly comment on this in the Discussion by adding the following text:

Our simulations and the results in figures 4 and 5 were obtained with a HiSeq model with a vpSOC(2) for the first two targets. We chose this because we found the complement configuration (i.e. vpSOC(1) + vpSOC(2)) to provide a poor match to the observed halfway coarticulation on the second segment, and the curvatures around both targets. From these results, we inferred that those movements that are hierarchical must be dynamically linked at the second target.

2.4.6 Model Complexity

Please report the total number of free parameters for each model. This information is essential to evaluate whether the increased flexibility of the HiSeq model could be explained simply by a greater number of parameters.

We thank the reviewer for this comment. We agree this is important information. We now report the total number of free parameters of both models, as well as the parameters we varied in our simulations. Additionally, we address model complexity in the Discussion. Concretely, we have added the following texts:

1. In the Models section of the Results, we now include the total number of parameters and those changed during simulations, for each model separately. Concretely:

The number of parameters for vpSOC(3) is 17, and for HiSeq 20. However, most of these parameters are determined by the task, or were fixed identically across both models in all simulations. This includes the three parameters unique to HiSeq (*monitor_distance*, *monitor_sd* and τ_{23}), which were set to match the target timing observed in participant trajectories. Note that vpSOC(N) can in principle include separate timing parameters for each segment, adding up to N free parameters per sequence. In our simulations, however, we fixed these timings as equally spread (just as we fixed the timing-related parameters in HiSeq), so we did not treat them as free parameters in vpSOC(3).

What matters more than the total number of parameters of each model, is how many parameters we varied to explain the behavioral data. The explanatory power of each model, as seen in Figure 4 and Figure 5, was obtained by changing 3 parameters for both models: r , h_1 and h_2 .

2. In the Results:

As can be clearly seen in Figure 4, HiSeq covers a much wider area than vpSOC(3), including more of our participant data: For instance, multiple participant-sequence combinations show a much higher curvature for highly-negatively coarticulated trajectories (i.e. those between x-values of -20 to -10 within the green area but not the black area). This greater coverage is not a trivial consequence of model complexity: both models were simulated by varying the same three parameters. Thus, the additional flexibility of HiSeq reflects its hierarchical structure rather than free parameterization. To ensure that this comparison does not bias results toward HiSeq, we combine the coverage analysis with a significance test: trajectories are considered incompatible with vpSOC(3) only when they lie outside its coverage and show coarticulation significantly different from zero. This controls for motor noise and confirms that the flat planning model has fundamental limitations.

3. In the “Simulation parameters” section of Methods:

The number of parameters for vpSOC(3) is 17, and for HiSeq 20. However, most of these parameters were determined by the task and the plant, or were fixed and equal for both models throughout all our simulations, including the two parameters unique to HiSeq (*monitor_sd* and *tau₂₃*), which were set to match the timing of targets observed in participant trajectories. Note that this timing can also be changed in vpSOC(N), possibly adding up to N parameters per sequence. As with HiSeq, we set the timings as equally spread, and therefore do not consider this a parameter of vpSOC(3) in our simulations. More important than the total number of parameters of each model, is how many parameters we varied to explain all the behavioral data. The explanatory power of each model, as seen in Figure 4 and Figure 5, was obtained by changing 3 parameters for both models: *r*, *h₁* for both models, and *monitor_distance* for HiSeq and *h₂* for vpSOC(3).

2.5 Minor Comments

- *There are several minor errors throughout the text and figure captions. For example: o The term “hierarchical” is omitted when referring to the HiSeq model (line 104). o Degrees of freedom are missing for the reported t-statistics (line 243). o Some figure panel labels appear to be incorrect or missing—for instance, the caption for Figure 4 refers to panel IDs that do not exist.*

We thank the reviewer for pointing out these errors. We have addressed them in the manuscript. In particular, we have ensured all the necessary information is included in all statistical tests, in addition to that pointed out by the reviewer.

2.6 Optimal feedback control

Optimal feedback control has been conceptualized as comprising multiple components—such as a feedback controller, forward model, and state estimator—each of which has been proposed to map onto distinct brain regions arranged hierarchically (Shadmehr & Krakauer, Exp Brain Res, 2008). From this perspective, the vpSOC model could already be considered hierarchical. Could the authors clarify in what specific sense they regard the vpSOC model as not hierarchical?

We have now added a clarification regarding the hierarchy to which we refer to when talking about the models. In short, we mean specifically a hierarchy of controllers. We have now added the following text to the Introduction (surrounding text included, with new text highlighted):

Body movements are known to be hierarchically **controlled** in the brain. Evidence for such a hierarchy **of controllers** has been found in neuroimaging and lesion studies of animals and humans, where simple, often short and single-target movements have been found to be learned and executed within the primary motor cortex (Yokoi and Diedrichsen, 2019; Kawai et al., 2015; Overduin et al., 2012), while the execution of complex movements has been found in pre-motor and parietal areas (Yokoi et al., 2018; Yokoi and Diedrichsen, 2019), and learning in basal ganglia (Tremblay et al., 2009) and the supplementary motor area (Verwey et al., 2002). **In this hierarchy, the simple controllers in the primary motor cortex could be deployed by pre-motor and parietal areas to create sequences of these single-target movements.**

2.7 Distinction between flat and hierarchical

Regarding the distinction between the “flat” and hierarchical models: the vpSOC(3) model plans all three movements simultaneously, while the HiSeq model consists of three vpSOC(1) controllers whose outputs are concatenated via SHC dynamics. Conceptually, could these two approaches be considered functionally similar in terms of planning? Clarification on how these models differ with respect to the hierarchical structure of control would be helpful.

Certainly. In response to a previous comment, we added the differences in planning between the models to the introduction. We include the changes here again for convenience:

We used two quantitative measures of coarticulation in combination with computational models to differentiate between flat (i.e. non-hierarchical) and hierarchical planning, **both reflecting a combination of anticipatory (Hansen et al., 2015) and carry-over (Hansen et al., 2018) coarticulation.** . To do this, we model the two planning strategies with two closely matched models. Both models are grounded in the same optimization framework, the stochastic optimal controller (SOC) (Todorov, 2005), and use identical movement primitives based on via-point control. The key difference lies in their architecture: in the first, a flat model plans the full sequence as a single, unified trajectory. In the second, a hierarchical model introduces a minimal and neurobiologically plausible control layer that dynamically links simpler movement segments. This extension allows the hierarchical model to flexibly compose multi-target sequences in real time without altering the underlying control principles. **Importantly, both models can plan movements in their entirety and differ only in which elements are required before movement onset: vpSOC requires all targets and control parameters to be planned before movement onset, consistent with the concept of chunking (Ramkumar et al., 2016) and competitive queuing (e.g. Yewbrey et al., 2023), while HiSeq can start by planning the first target(s) and add more as needed, consistent with recent observations regarding planning horizons in a similar sequential task (Kashefi et al., 2024).** . By holding the core control principles and simple movements constant across models, we ensure that observed differences in fit arise from the presence of hierarchical structure, and not simply from a richer set of low-level parameters.

3 Reviewer 3

The manuscript addresses an interesting and relevant question in the current landscape of movement neuroscience, that is, how sequences of movements are produced and controlled by the brain. The experiment is overall well designed, but the metrics selected to assess participants' kinematics and evaluate coarticulation lack the conceptual grounding to assess the phenomenon at hand.

3.1 Comment 1

The key question that we are trying to answer when we assess coarticulation is: given a sequence of movements (e.g., reaching movements like in this case), are the movement parameters extracted from segment n able to predict segment $n+k$? Or, in a more nuanced version, how early the movement parameters extracted from segment n become predictive of segment $n+k$? Note that I am using $n+k$ rather than $n+1$ because coarticulation can span further the next movement and cover a longer chunk of the sequence. I encourage the authors to rethink the metrics they adopted by considering this conceptual framework. For example, in MoveID 0, 1, 2, which all start with reaching target 1, is the first segment predictive of the second target? Having answered this participant-wise, they will have a compact measure of coarticulation, that they can now use in their models. The current metrics make the results very hard to interpret in terms of coarticulation.

We appreciate the reviewer's emphasis on a prediction-based conceptual framework for assessing coarticulation. This perspective, focused on how movement parameters in one segment inform the following segments, has been highly successful in experimental research. Indeed, our modeling approach builds directly on this logic: both vpSOC and HiSeq explicitly generate predictions for upcoming segments, and the degree to which these predictions affect kinematics is precisely what our measures (halfway and transitional coarticulation) aim to capture. In this sense, we see our analysis as complementary to the reviewer's proposed $n+k$ predictive framework. Our contribution operationalizes similar anticipatory relationships, but embeds them within a model-based comparison to differentiate flat versus hierarchical sources of prediction. We fully agree that more explicit predictive metrics are valuable, and we consider this a promising direction for future expansions of our approach based on generative models of coarticulation. To make this focus clearer, we added the following text (in red) to the Introduction:

Importantly, and in contrast to standard experiments, we designed the task using large target circles to foster coarticulation at the kinematic level. This enabled us to observe and directly model coarticulation. **Coarticulation itself can be seen as a sign of planning ahead, as it can be predictive of the next target in a sequence, as is the case in our experiment. However, as both planning strategies (flat and hierarchical) can produce trajectories with coarticulation predictive of the next target, coarticulation is not, by itself, sufficient to infer the hierarchical controllers. To account for this**, we focused on the sign of coarticulation, and the maximum curvature as trajectories transitioned from one target to the next. The two features coarticulation and curvature are important because their relationship reveals how the movement was planned. If the entire sequence is generated as a single movement (flat planning), these two features are tightly linked. But if the sequence is composed by dynamically linking simpler movements (hierarchical planning), they can vary more independently.

3.1.1 Curvature and coarticulation

The curvature of the trajectory, or how much the participant cuts the corner at a via point are to an extent epiphenomenal to coarticulation, but per se do not allow to conclude that there is coarticulation.

We agree with the reviewer that corner cutting itself is not necessarily coarticulation. However, we would like to clarify that curvature (i.e. the maximum curvature around a target, as plotted in Figure 4) is not the same as transitional coarticulation and is independent from coarticulation. We have now added the following text to the Results section:

1. When first defining the transitional crosses in Results, we have added the following text (new text highlighted):

In addition to halfway coarticulation, we found evidence of a second type of coarticulation as the movement transitioned from one target to the next. This can be observed as trajectories that do not go through the center of the target, instead having a distribution (over trials) that is centered elsewhere within the target circle. To analyze this, we determined for each trajectory separately the point at which it crossed a line that equally divides the angle between the center-to-center line to the first target and the center-to-center line from the first target to the second target (see Figure 8B in Methods). We call these the transitional crosses (TC). We define the transitional coarticulation tC_n as the mean of transitional crosses (TC) over all trials of the same sequence around target n. **As can be seen in Figure 3, the sequence-dependent distributions of the transitional crosses depend on the following target. This difference is the most pronounced between sequences 0 and 2 for both representative participants. Because of this, we believe it is a measure of a mixture of anticipatory and carry-over coarticulation**

2. When defining curvature, we now clarify how it relates to transitional coarticulation:

We now build on the correlations from the previous section to categorize participants' movements as being consistent with either flat or hierarchical planning. To do this, we analyze the maximum curvature of trajectories (see Methods) as they transition from one target to the next. **While lowering the maximum curvature around a target lead to coarticulation, it is not necessarily directly related to coarticulation. Indeed, as can be seen in Figure 4, curvature and coarticulation are not correlated. Instead, maximum curvature is more closely related to segment fusion, which quantifies how continuously two movement segments are executed. This relationship is captured by the fusion index (Sporn et al., 2022), where lower values correspond to higher curvature and thus more fused trajectories. Importantly, the usefulness of curvature as a measure will depend on the geometry of the task: if a sequence is to be performed with via-points lying on a straight line, maximum curvature will be near zero, while the fusion index could depend on the level of training the participant has.** We will show that HiSeq's disentanglement of halfway and transitional coarticulation matches what we observed in participant data for many trajectories. Using these measures, we specify two criteria to determine whether a participant trajectory is compatible only with hierarchical planning (HiSeq), only with non-hierarchical planning (vpSOC(3)), or with both.

3.2 Comment 2

Considering the above, it is unclear to me how halfway and transitional coarticulation are different “types” of coarticulation. What distinct aspects of the process of linking two subsequent movement segments do these metrics reflect? I am not convinced this comes across clearly in the manuscript, which makes the use of these two measures of coarticulation not fully justified.

We apologize for the confusion in this regard. For clarification, halfway coarticulation and transitional coarticulation refer not to two types of coarticulation, but two distinct measures of them. They both reflect a mixture of anticipatory and carry-over coarticulation at the kinematic level. We have added the following clarifications:

1. In the Introduction:

We used two quantitative measures of coarticulation in combination with computational models to differentiate between flat (i.e. non-hierarchical) and hierarchical planning, **both reflecting a combination of anticipatory (Hansen et al., 2015) and carry-over (Hansen et al., 2018) coarticulation.**

2. In Results, when defining the halfway coarticulation:

These deviations are a sign of coarticulation, as they depend not only on the current target, but also on the previous and next ones (if any); we therefore define the halfway coarticulation as the average HC over all 20 repetitions of the same sequence. **This measure reflects a combination of anticipatory and carry-over coarticulation, in movements that have a preceding target and/or a following target, respectively.**

3. In Results, when defining transitional coarticulation:

We define the transitional coarticulation tC_n as the mean of TCs over all trials of the same sequence around target n . **Like with halfway coarticulation, transitional coarticulation reflects both anticipatory effects of the upcoming target and carry-over effects of the preceding movement, but it is measured at the target rather than mid-movement between targets**

3.3 Comment 3

In terms of writing, I would also encourage the authors to reduce the amount of jargon and acronyms in the manuscript – as it is, the text is not easy to follow, and one needs to keep going through the glossary. For example, why call the sequences MoveID, which is very peculiar, and not just sequence 1, 2, etc? More in general, I suggest revising the manuscript in a way that reduces to a minimum the very need for a glossary.

We have now simplified the text by removing the least-used acronyms, as well as changing the naming conventions for the sequences (MoveID X \rightarrow Sequence X). Additionally, we now define the remaining acronyms at their first use in any section, to improve readability.

3.4 Comment 4

Some figures are also hard to read. For example, Figure 2 is clear with respect to the trajectories, whereas plotting velocities from all trials on top of each other makes individual traces very difficult to appreciate. I encourage the authors to plot the average velocity accompanied (if needed) by individual trials representative of features of interest they wish to highlight for the reader. Also related to visualisation, in Figure 3, the x axis reflects the sequence, which is now called Sq0, 1, 2 etc, instead of MoveID as before. I strongly suggest adopting common naming conventions.

We apologize for this oversight. We have now changed the visualization of the speed profiles in Figure 2 to better display their structure. Specifically, we have changed the speed profile plotting to a segment-based one, where each segment (i.e. from target k to $k+1$) is normalized in time for each trial separately. With this, the x-axis values of e.g. 0 to 1 represent the flight time of the first segment. We did this to account for variations in segment duration within a trajectory and across trials. We have additionally included the following description of this procedure in the Methods section:

To show the speed profiles in Figure 2, we split sequential trajectories in three segments, each one corresponding to one target. To find the points of separation between targets, we found the minimum speed of movement within each target, and split the trajectories into before-target and after-target based on the position of this minimum. We then normalized time for each segment, such that each segment had a duration of 1. The x-axis in the speed profiles of Figure 2 show this normalized time.

4 References

- Acuna, D. E., Wymbs, N. F., Reynolds, C. A., Picard, N., Turner, R. S., Strick, P. L., Grafton, S. T., and Kording, K. P. (2014). Multifaceted aspects of chunking enable robust algorithms. *Journal of Neurophysiology*, 112(8):1849–1856.
- Ariani, G. and Diedrichsen, J. (2019). Sequence learning is driven by improvements in motor planning. *Journal of Neurophysiology*.
- Crevecoeur, F., Thonnard, J.-L., and Lefèvre, P. (2020). A Very Fast Time Scale of Human Motor Adaptation: Within Movement Adjustments of Internal Representations during Reaching. *eNeuro*, 7(1).
- Cuevas Rivera, D., Strobel, A., Goschke, T., and Kiebel, S. J. (2020). Modeling dynamic allocation of effort in a sequential task using discounting models. *Frontiers in Neuroscience*, 14.
- Fine, J. M., Moore, D., and Santello, M. (2017). Neural oscillations reflect latent learning states underlying dual-context sensorimotor adaptation. *NeuroImage*, 163:93–105.
- Gelman, A., Hwang, J., and Vehtari, A. (2014). Understanding predictive information criteria for Bayesian models. *Statistics and Computing*, 24(6):997–1016.
- Guigon, E., Chafik, O., Jarrassé, N., and Roby-Brami, A. (2019). Experimental and theoretical study of velocity fluctuations during slow movements in humans. *Journal of Neurophysiology*, 121(2):715–727.
- Hansen, E., Grimme, B., Reimann, H., and Schöner, G. (2015). Carry-over coarticulation in joint angles. *Experimental Brain Research*, 233(9):2555–2569.
- Hansen, E., Grimme, B., Reimann, H., and Schöner, G. (2018). Anticipatory coarticulation in non-speeded arm movements can be motor-equivalent, carry-over coarticulation always is. *Experimental Brain Research*, 236(5):1293–1307.
- Kashefi, M., Reschechtko, S., Ariani, G., Shahbazi, M., Tan, A., Diedrichsen, J., and Pruszynski, J. A. (2024). Future movement plans interact in sequential arm movements. *eLife*, 13:RP94485.
- Kawai, R., Markman, T., Poddar, R., Ko, R., Fantana, A. L., Dhawale, A. K., Kampff, A. R., and Ölveczky, B. P. (2015). Motor Cortex Is Required for Learning but Not for Executing a Motor Skill. *Neuron*, 86(3):800–812.
- Kornysheva, K., Bush, D., Meyer, S. S., Sadnicka, A., Barnes, G., and Burgess, N. (2019). Neural Competitive Queuing of Ordinal Structure Underlies Skilled Sequential Action. *Neuron*, 101(6):1166–1180.e3.
- Legler, E., Cuevas Rivera, D., Schwöbel, S., Wagner, B. J., and Kiebel, S. (2025). Cognitive computational model reveals repetition bias in a sequential decision-making task. *Communications Psychology*, 3(1):92.
- Mantziara, M., Ivanov, T., Houghton, G., and Kornysheva, K. (2021). Competitive state of movements during planning predicts sequence performance. *Journal of Neurophysiology*, 125(4):1251–1268.
- Overduin, S. A., d’Avella, A., Carmena, J. M., and Bizzi, E. (2012). Microstimulation Activates a Handful of Muscle Synergies. *Neuron*, 76(6):1071–1077.
- Ramkumar, P., Acuna, D. E., Berniker, M., Grafton, S. T., Turner, R. S., and Kording, K. P. (2016). Chunking as the result of an efficiency computation trade-off. *Nature Communications*, 7(1):12176.

- Rand, M. K., Alberts, J. L., Stelmach, G. E., and Bloedel, J. R. (1997). The influence of movement segment difficulty on movements with two-stroke sequence. *Experimental Brain Research*, 115(1):137–146.
- Rand, M. K. and Stelmach, G. E. (2000). Segment interdependency and difficulty in two-stroke sequences. *Experimental Brain Research*, 134(2):228–236.
- Sakai, K., Kitaguchi, K., and Hikosaka, O. (2003). Chunking during human visuomotor sequence learning. *Experimental Brain Research*, 152(2):229–242.
- Sosnik, R., Flash, T., Hauptmann, B., and Karni, A. (2007). The acquisition and implementation of the smoothness maximization motion strategy is dependent on spatial accuracy demands. *Experimental Brain Research*, 176(2):311–331.
- Sosnik, R., Hauptmann, B., Karni, A., and Flash, T. (2004). When practice leads to co-articulation: The evolution of geometrically defined movement primitives. *Experimental Brain Research*, 156(4):422–438.
- Sporn, S., Chen, X., and Galea, J. M. (2022). The dissociable effects of reward on sequential motor behavior. *Journal of Neurophysiology*, 128(1):86–104.
- Todorov, E. (2005). Stochastic Optimal Control and Estimation Methods Adapted to the Noise Characteristics of the Sensorimotor System. *Neural Computation*, 17(5):1084–1108.
- Todorov, E. and Jordan, M. I. (1998). Smoothness Maximization Along a Predefined Path Accurately Predicts the Speed Profiles of Complex Arm Movements. *Journal of Neurophysiology*, 80(2):696–714.
- Tremblay, P.-L., Bedard, M.-A., Levesque, M., Chebli, M., Parent, M., Courtemanche, R., and Blanchet, P. J. (2009). Motor sequence learning in primate: Role of the D2 receptor in movement chunking during consolidation. *Behavioural Brain Research*, 198(1):231–239.
- Verwey, W. B., Abrahamse, E. L., Ruitenberg, M. F. L., Jiménez, L., and de Kleine, E. (2011). Motor skill learning in the middle-aged: Limited development of motor chunks and explicit sequence knowledge. *Psychological Research*, 75(5):406–422.
- Verwey, W. B., Lammens, R., and van Honk, J. (2002). On the role of the SMA in the discrete sequence production task: A TMS study. *Neuropsychologia*, 40(8):1268–1276.
- Wymbs, N. F., Bassett, D. S., Mucha, P. J., Porter, M. A., and Grafton, S. T. (2012). Differential Recruitment of the Sensorimotor Putamen and Frontoparietal Cortex during Motor Chunking in Humans. *Neuron*, 74(5):936–946.
- Yeo, S.-H., Franklin, D. W., and Wolpert, D. M. (2016). When Optimal Feedback Control Is Not Enough: Feedforward Strategies Are Required for Optimal Control with Active Sensing. *PLoS Computational Biology*, 12(12):e1005190.
- Yewbrey, R., Mantziara, M., and Kornysheva, K. (2023). Cortical Patterns Shift from Sequence Feature Separation during Planning to Integration during Motor Execution. *Journal of Neuroscience*, 43(10):1742–1756.
- Yokoi, A., Arbuckle, S. A., and Diedrichsen, J. (2018). The Role of Human Primary Motor Cortex in the Production of Skilled Finger Sequences. *Journal of Neuroscience*, 38(6):1430–1442.
- Yokoi, A. and Diedrichsen, J. (2019). Neural Organization of Hierarchical Motor Sequence Representations in the Human Neocortex. *Neuron*, 103(6):1178–1190.e7.

Response to reviewers

Dario Cuevas Rivera, Stefan Kiebel

February 2, 2026

1 Editor

1.1

As part of the editorial requests, we have added the following text in the Introduction stating how we think our work is placed within the literature:

Taken together, this body of work indicates that behavioral evidence for hierarchical control has so far relied primarily on indirect timing- and error-based measures that are strongly influenced by training. This makes it difficult to dissociate changes in planning structure from more general improvements in motor execution. What is therefore missing is a way to probe hierarchical organization at the level of movement generation itself, without relying on long training protocols or reaction-time proxies.

1.2

Additionally, we have added explicit hypotheses that our work was meant to test: Concisely, the main objective in this work is to show that a flat planning strategy does not fit all trajectories observed in our experiment. We therefore hypothesized that many participant sequential trajectories would show geometries that cannot be produced with vpSOC alone. Concisely, that the types and magnitude of coarticulation would vary between segments of the same sequential movement, contrary to the assumptions of the non-hierarchical (i.e. flat) optimal control model. We further hypothesized that many of these trajectories would be consistent with a hierarchical planning strategy, given the hierarchical model's ability to mix movement segments with different properties.

Consistent with our hypotheses, we found many participant sequences in which coarticulation and smoothness were outside of the range that can be produced with vpSOC. We further showed that trajectories were consistent with a hierarchical strategy, where the first two targets are grouped into one movement, and the third one is reached individually, with its trajectory dynamically linked to the initial two-target movement. Further analyses revealed a link between the hierarchical nature of some of the observed trajectories, and existing measures related to chunking, namely the fusion index.

2 Reviewer 1

2.1

Here, the authors state "However, how these measures are related to the hierarchical organization of movements is still unclear." but providing one or two concrete examples would help clarify and strengthen this point. Although the authors mentioned "Furthermore, these studies have used extensive training to observe their results, which has been shown to reduce reaction times, error rates, inter-movement intervals, and movement fluidity even for untrained sequences [...], affecting results based on these measures." later in the text, it would be helpful to explain more explicitly why this is considered problematic. For instance, is the concern that behavioral measures of hierarchical structure in the trained sequences might be obscured by an apparent improvement in an untrained sequences, or is a more fundamental issue being suggested? Clarifying this would improve the reader's understanding of the authors' argument.

We have revised the text in the introduction to explain why these indirect measures of chunking and hierarchical organization are problematic. Concisely, we have updated the following texts:

Chunking [8] has been proposed as a mechanism underlying, at least partially, the hierarchical organization of movements, whereby individual movements that are executed in a sequence many times are grouped together in the brain and become a single action [7]. As such, chunking could be the result of the flattening of the hierarchical controller that was originally needed to perform a sequential action, simplifying its further execution.

...

However, how these measures are related to the hierarchical organization of movements is still unclear: for instance, chunking is thought of as simplifying the control hierarchy (i.e. flattening), but could also be the result of a fine-tuning of the parameters of this hierarchy, leading to skilled performance after training. Similarly, pre-planning could be associated with both a hierarchical and a flat control strategy.

...

Furthermore, these studies have used extensive training to observe their results, which has been shown to reduce reaction times, error rates, inter-movement intervals, and movement fluidity even for untrained sequences [1, 3, 5, 9]. Because untrained sequences also show these improvements in performance, it is unclear how much the improvement can be attributed to practicing a sequence in general, and chunking of that sequence in particular.

2.2

Also, after reading the manuscript several times, the claim of “direct evidence” of hierarchical structure still appears somewhat strong. In the current form, the hierarchical structure seems to be inferred from a fixed set of computational models, with model selection based on comparing the range of predicted kinematic indices across models against the observed data. This approach is informative but remains relatively indirect. While the comparison may rule out flat model (vpSOC), it does not necessarily establish that the alternative hierarchical model (HiSeq) is the correct account. To strengthen this point, the authors may wish to either temper the wording of the “direct evidence” claim or provide additional analyses or data. For example, fitting the free parameters of the HiSeq model and examining whether a single parameter set (or at least a constrained subset of parameters) can reliably reproduce the observed data would provide stronger support for the proposed interpretation. In addition, the authors may consider whether the model yields testable predictions that could be examined in an independent experiment (e.g., changing target sizes and arrangements). Demonstrating that a novel prediction derived from the model is fulfilled would provide particularly strong support for the claim of hierarchical structure without additional training and would substantially strengthen the interpretation as “direct evidence.” Even if such an experiment is beyond the scope of the current study, clarifying these predictions would help delineate the model’s explanatory power.

We agree with the reviewer that the evidence produced by our methods is not direct. To this effect, we have revised parts of the introduction (listed below). However, as we have now clarified in the Introduction (see below) and in the Discussion (below), the main goal of our work was to show that the flat strategy cannot reproduce the experimentally-observed variability in trajectories; HiSeq was introduced to show how a hierarchical controller could disentangle curvatures and coarticulation in a very direct way. However, as was argued by [4], the simplicity of the model, in particular of the higher level in the hierarchy, makes model fitting and predictions meaningless. For these reasons, we agree with the reviewer that we do not establish HiSeq as the correct account and have therefore revised the wording surrounding this issue.

Below are the revised texts from the Introduction:

In this work, we introduce a rapid and training-free approach to obtain behavioral evidence of the hierarchical structure in human reaching movements. Our model-based method does not rely on secondary behavioral measures, such as reaction times or smoothness, nor on comparing early and late trials. Instead, it infers the presence of hierarchical control based directly on the geometry of the produced trajectories.

...

Concisely, the main objective in this work is to show that a flat planning strategy does not fit all trajectories observed in our experiment. We therefore hypothesized that many participant sequential trajectories would show geometries that cannot be produced with vpSOC alone. Concisely, that the types and magnitude of coarticulation would vary between segments of the same sequential movement, contrary to the assumptions of the non-hierarchical (i.e. flat) optimal control model. We further hypothesized that many of these trajectories would be consistent with a hierarchical

planning strategy, given the hierarchical model’s ability to mix movement segments with different properties.

...

Consistent with our hypotheses, we found many participant sequences in which coarticulation and smoothness were outside of the range that can be produced with vpSOC. We further showed that trajectories were consistent with a hierarchical strategy, where the first two targets are grouped into one movement, and the third one is reached individually, with its trajectory dynamically linked to the initial two-target movement. Further analyses revealed a link between the hierarchical nature of some of the observed trajectories, and existing measures related to chunking, namely the fusion index.

New text in Discussion-Limitations:

Finally, our main result is to show that a flat controller is not capable of displaying the wealth of trajectories observed in our experiment. We did this by showing that the state-of-the-art model based on optimal control could not reproduce many of the trajectories. We introduced HiSeq as the simplest extension of vpSOC to a hierarchical controller which is capable of not only linking individual components (e.g. vpSOC(2) and vpSOC(1)), but also of creating coarticulation at the transition between the two. The choice of the generalized Lotka-Volterra equations as the top layer was based on their previous use in neuronal dynamics [6, 2], but we did not specifically test their viability in this application.

3 Reviewer 3

3.1

In the new version of the introduction, the authors claim that “how these measures [e.g., chunking and anticipatory planning] are related to the hierarchical organization of movements is still unclear”. In what way it is not clear? To me, chunking looks very much like a proxy of hierarchical movement control. In other words, it is not clear how the model-based approach proposed here unveils the hierarchical structure of voluntary movement more effectively than chunking.

We have revised the Introduction to clarify why measures of chunking might not directly reveal the presence or absence of hierarchical control of sequential movements. Below are the revised texts from the Introduction:

Chunking [8] has been proposed as a mechanism underlying, at least partially, the hierarchical organization of movements, whereby individual movements that are executed in a sequence many times are grouped together in the brain and become a single action [7]. As such, chunking could be the result of the flattening of the hierarchical controller that was originally needed to perform a sequential action, simplifying its further execution.

...

However, how these measures are related to the hierarchical organization of movements is still unclear: for instance, chunking is thought of as simplifying the control hierarchy (i.e. flattening), but could also be the result of a fine-tuning of the parameters of this hierarchy, leading to skilled performance after training. Similarly, pre-planning could be associated with both a hierarchical and a flat control strategy.

3.2

Subsequently, the authors argue that “these [previous] studies have used extensive training to observe their results, which has been shown to reduce reaction times, error rates, inter-movement intervals, and movement fluidity even for untrained sequences, affecting results based on these measures.” But still, in spite of these issues, previous work has nevertheless provided evidence of the hierarchical organisation of voluntary movement. I can see the practical advantage of a paradigm that requires shorter or no training, but what new discovery about how the brain controls movements does this new approach foster?

We have now revised the Introduction to clarify why chunking might not be directly related to hierarchical control:

Chunking [8] has been proposed as a mechanism underlying, at least partially, the hierarchical organization of movements, whereby individual movements that are executed in a sequence many times are grouped together in the brain and become a single action [7]. As such, chunking could

be the result of the flattening of the hierarchical controller that was originally needed to perform a sequential action, simplifying its further execution.

...

However, how these measures are related to the hierarchical organization of movements is still unclear: for instance, chunking is thought of as simplifying the control hierarchy (i.e. flattening), but could also be the result of a fine-tuning of the parameters of this hierarchy, leading to skilled performance after training. Similarly, pre-planning could be associated with both a hierarchical and a flat control strategy.